**Subject Category:**
Biology (whole organism)

ecology/evolution/behaviour

Atlantic salmon, hydropower, smolt, migration, selection, survival

**Author for correspondence:**
Tormod Haraldstad
e-mail: tormod.haraldstad@niva.no

# Migratory passage structures at hydropower plants as potential physiological and behavioural selective agents

Tormod Haraldstad[1,2], Thrond Oddvar Haugen[3], Frode Kroglund[4], Esben Moland Olsen[2,5] and Erik Höglund[1,2]

[1]Norwegian Institute for Water Research (NIVA), N-4879 Grimstad, Norway
[2]Centre for Coastal Research, University of Agder, N-4604 Kristiansand, Norway
[3]Faculty of Environmental Sciences and Natural Resource Management, Norwegian University of Life Sciences, NO-1432 Ås, Norway
[4]County Governor of Aust- and Vest-Agder, N-4809 Arendal, Norway
[5]Institute of Marine Research (IMR), Flødevigen, N-4817 His, Norway

TH, 0000-0002-9575-5144; EH, 0000-0002-1350-8255

Anthropogenic activities affect fish populations worldwide. River dams have profound impacts on ecosystems by changing habitats and hindering migration. In an effort to counteract such effects, a range of mitigation measures have been installed at hydroelectric power plants. However, not all individuals in a population use these measures, potentially creating strong selection processes at hydroelectric power plants. This may be especially true during migration; fish can get heavily delayed or pass through a hydropower turbine, thus facing increased mortality compared with those using a safe bypass route. In this study, we quantify migration route choices of descending wild passive integrated transponder (PIT)-tagged Atlantic salmon smolts released upstream from a hydroelectric plant. We demonstrate how only a few metres' displacement of bypass canals can have a large impact on the fish guidance efficiency (FGE). The proportion of fish using the bypasses increased from 1% to 34% when water was released in surface gates closer to the turbine intake. During a period of low FGE, we observed two different smolt migratory strategies. While some individuals spent little time in the forebay before migrating through the turbine tunnel, others remained there. We suggest that these groups represent different behavioural types, and that suboptimal mitigation measures at hydropower intakes may, therefore, induce strong selection on salmon behavioural traits. The ultimate outcome of these selection mechanisms is

discussed in light of potential trade-offs between turbine migration mortality coast and optimal sea entrance timing survival benefits.

# 1. Introduction

By reducing river connectivity and thereby blocking or slowing down fish migration, hydropower dams are considered one of the main challenges for restoring and maintaining sustainable fish populations worldwide [1,2]. To complete an anadromous or catadromous life cycle, fish require unimpeded migration routes between freshwater and seawater, for both descending and ascending migrants [3], and a range of mitigation measures have been tried to address this problem [4].

Mitigation measures need to be appropriately aligned to the individual location and the specific behaviour of the targeted species. For instance, downstream migrating salmonid smolts are mainly surface orientated and follow the main river flow. Thus, mitigation measures for Atlantic salmon (*Salmo salar*) smolts are adjusted to this behaviour and guide fish away from the turbine inlet towards a safe bypass and further downstream [5]. The guidance structures can be mechanical barriers that prevent fish from entering hazardous areas or behavioural barriers, repelling fish from hazardous area and/or guiding fish towards a safe area. When aggregated in a safe area, mechanical fish collection systems remove and transport fish further downstream; alternatively, fish swim past the obstacle and into the tailrace via bypass channel systems. However, despite good intentions, some mitigation measures are inefficient or only benefit a part of the population [6,7]. Moreover, since the migration delay and turbine passing are both associated with mortality [4,8], there are potentially strong selection processes at hydropower intakes.

There is mounting evidence that human impacts on wild animal populations are not limited to ecological changes but may also involve strong directional selection and contemporary evolutionary changes [9]. In particular, harvest-induced selection and evolution of life-history traits, such as growth and maturation, have received much attention [10], while fewer studies have investigated human-induced selection and evolutionary change of animal behavioural traits [11]. Despite being a global threat to freshwater fish migration and therefore population viabilities, hydropower-induced selection has so far attracted minimal attention (but see [12,13]).

The smolt run of Atlantic salmon is a fine-tuned migration event, where the majority of a cohort leave their natal river during a period of a few weeks to start their migration towards the feeding areas in the North Atlantic Ocean [14]. The migration of physiologically prepared smolt is initiated by environmental cues in the river, such as changes in temperature or flow [15,16], that coincide with optimal temperature and food supply in the coastal areas [17]. Due to the physiological sensitivity and high predation risk of smolt and post-smolt individuals, these are critical stages in the life cycle of salmonids [15]. Entering saltwater at the right time is essential for survival, and this period of optimal environmental conditions is often termed the smolt window [18]. In general, heavy delay of Atlantic salmon smolt migration is likely to have highly negative impact on survival. The delayed smolt may suffer from increased predation and accumulated energetic costs [18–20].

Damming of rivers may affect both the environmental cues that initiate the smolt run and alter the timing of sea entrance [21]. Water reservoirs in the mountain areas have the capacity to withhold a large amount of water during high precipitation periods and conversely release water during droughts. The natural discharge pattern in the downstream rivers is thus flattened out and controlled by hydropower production profitability rather than natural precipitation variations and catchment run-off. In addition, retention of water from the higher altitude catchment areas may alter river temperatures downstream. By holding back meltwater in spring, these rivers are dominated by low-altitude tributary run-off with higher temperature rather than a mixture of the two. In addition, general lower river discharges cause a faster temperature impact from the external environment throughout the year. Due to hydropower-induced changes in river temperatures, cues that initiate smolt run timing are altered and smolts may not reach the coastal waters when food and temperature are optimal for survival.

Fish migrating through a hydropower turbine are associated with negatively size-dependent mortality [4,8], while using a safe bypass secures survival. Turbine intakes are typically covered by metal gratings or 'trash racks'. These are often substantially submerged, shaded and with higher water flow than the close-to-surface bypass alternatives. Therefore, individual smolt must choose between passage alternatives with very different properties potentially involving individual behaviour and physiological characteristics related to personalities and swimming capacity. Such selection

processes may be crucial in river systems where mitigation measures ought to be timed with phenological events, such as the smolting in salmonids. Thus, the knowledge about the overall efficiency and consequences of possible selection processes at fish passage facilities is needed for optimizing survival and mitigating hydropower-induced selection on behaviour traits.

In this study, we quantify the migration behaviour of wild passive integrated transponder (PIT)-tagged Atlantic salmon smolts released upstream of a newly built hydroelectric plant (HEP). We tested the hypothesis that the placement of a bypass (distance) in relation to the turbine intake (i.e. the distance between the two) is a proxy for its guidance efficiency. In addition, we examine if the trash rack (50 mm spacing) will function as a behaviour barrier, causing repellent behaviour for downstream migrating smolts. Furthermore, we discuss to what extent this repellent effect could be related to fish behaviour characteristics.

## 2. Methods

The River Storelva, Norway (58°40′9.99 N, 8°58′48.99 E; figure 1) has been regulated for hydroelectric power production since 2008. Fosstveit HEP is a run-of-the-river plant located 6 km above the river mouth and is the only HEP in the catchment area. It uses a 14.5 m high waterfall and the power-generating water comes from a small river forebay through one Kaplan turbine (4 blades, 330 r.p.m.) that is led back into the river through a tunnel tailrace, leaving a residual flow stretch of 230 m between the dam and the downstream tunnel tailrace. At the tunnel inlet, there is a 25 m$^2$ conventional trash rack with 50 mm spacing between the vertical bars mounted at a 70° angle from the vertical. A concrete wall covers the uppermost 0.5 m to avoid icing on the rack during winter. At the hydropower dam, there are four surface spill gates (trash gate, fish ladder and two floodgates) that may be used as safe bypasses past the hydropower facility for descending smolts. During the smolt run period, the gates were opened at different time intervals (table 1). In general, the water velocity at the trash gate in front of the tunnel inlet area varies with river discharge. If the power plant uses less than 16 m$^3$ s$^{-1}$ ($Q_{max}$), the water velocity never exceeds 0.64 m s$^{-1}$ ($Q_{max}$/rack area).

Wild Atlantic salmon smolts were caught in the uppermost rotary screw trap (RST; figure 1) on their downstream migration [22]. The smolts were anaesthetized with benzocaine (40–50 mg l$^{-1}$, ACD Pharmaceuticals AS) before being tagged internally with 23 mm PIT tags (23 mm, half-duplex, Oregon RFID). The tags were inserted through a small incision made ventrally between the posterior tip of the pectoral fin and the anterior point of the pelvic girdle. Based on the previous findings, the incision closed and healed without suturing [23]. The tagged fish were held for one day before being released at the catch site 350 m upstream the hydropower dam. A total of 923 smolts were released between 30 April and 21 May 2010. Migrating smolt could move past the dam using either the turbine tunnel or one of the four surface gates. The turbine migration route was open throughout the smolt run, while the opening of the different surface gates was alternated for the purpose of the experiment. The surface gates were opened sequentially, starting with the gate farthest away from the turbine followed by the one closer to the turbine (table 1). In addition, the fish ladder was opened again from 18 to 19 May to allow upstream migration of Atlantic salmon and sea trout (*Salmo trutta*) spawners that aggregated downstream of the dam. This is in accordance with the concession to operate, which is required of a Norwegian hydropower plant and includes site-specific compensation measures to mitigate possible damage caused to the environment.

The smolts were detected at three PIT antennas (TIRIS RI-CTL MB2A; Oregon RFID) and three RSTs between release site and river mouth. Smolts, using one of the surface gates in the hydropower dam, were detected in a PIT antenna in the residual flow stretch between the dam and the turbine tailrace (figure 1). The detection probability for this PIT antenna was estimated to be 100%, and detection in this antenna was used as evidence for migration through one of the surface gates [24]. An RST with leader net caught the turbine migrants, both dead and alive, in the tailrace (catch probability: 62%). In addition, both turbine and surface gate migrants could be detected in three PIT antennas (catch probability: 45% and 79%) and two RSTs (catch probability: 32% and 21%) between Fosstveit and the river mouth [24]. Only detections in the antennas and traps at Fosstveit were used for the estimation of forebay time (time-to-event analysis) to avoid including time spent in the river stretch between Fosstveit HEP and recapture location further downstream.

The estimation of detection probability for the PIT antenna in the residual flow stretch between the dam and the turbine tailrace was based on the detections of tagged smolts released upstream of the antenna on five different occasions ($n = 50$). This antenna covered the total water column and a

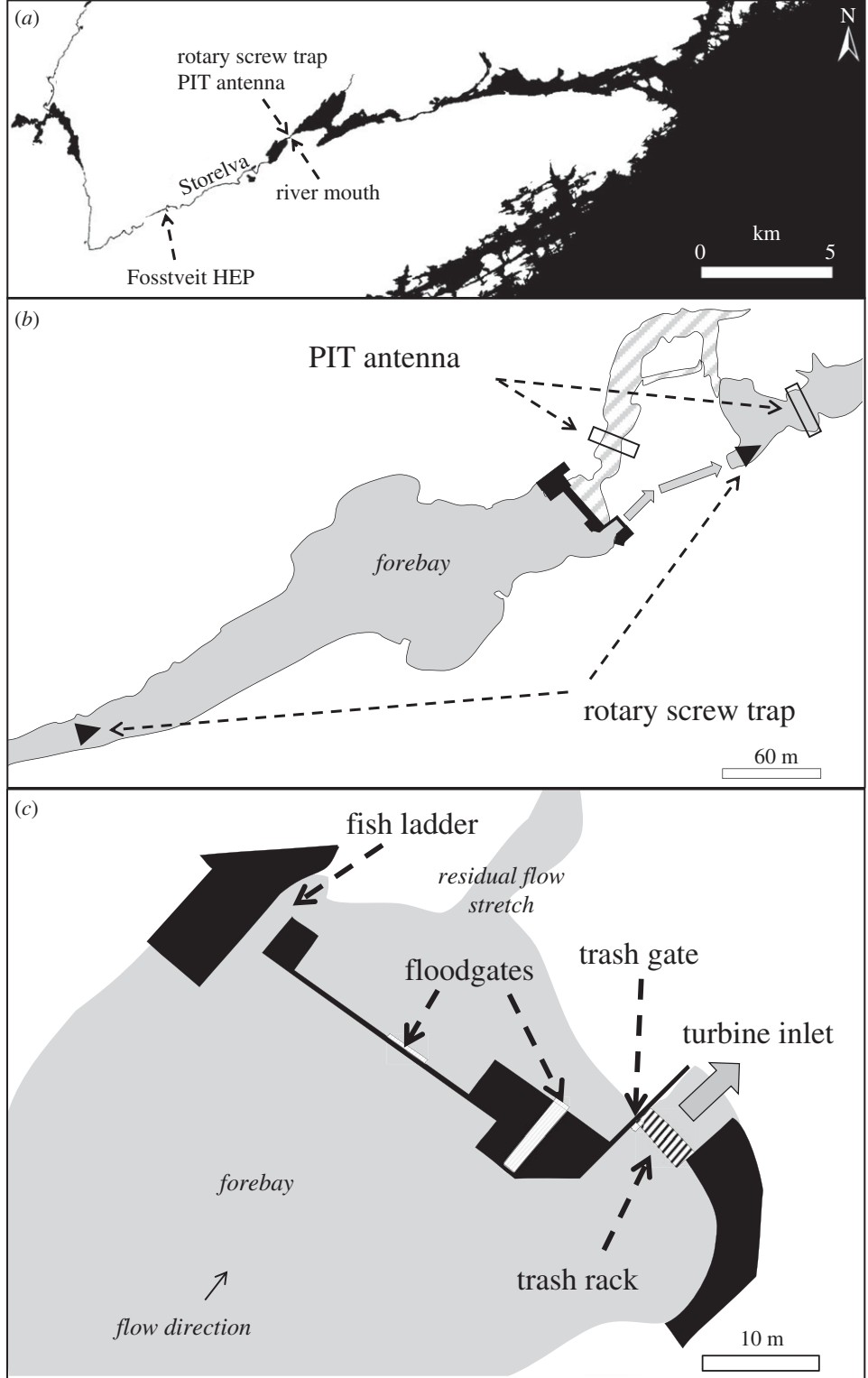

**Figure 1.** (*a*) The anadromous parts of River Storelva. (*b*) Fosstveit hydropower station with forebay, residual flow stretch (reduced discharge) and tailrace including catch and recapture locations. (*c*) Schematic diagram of Fosstveit dam with intake trash rack and different surface gates.

higher detection probability at this antenna, compared with the others, is expected due to the reduced discharge in this river stretch. The other PIT antenna detection probability ($p_{PIT}$) and catchability of RST ($p_{RST}$) were estimated from the mark-recapture analysis in program MARK [25], by fitting the sequential Cormack–Jolly–Seber model [26] to the individual recapture histories (see [24] for details).

**Table 1.** Experimental design where the different gates at Fosstveit hydropower dam with individual opening days (grey shades), size and distance to turbine inlet and associated river discharge and temperature (May 2010).

| | size (m) (width × depth) | distance to turbine inlet (m) | May | | | | | |
|---|---|---|---|---|---|---|---|---|
| | | | 1–3 | 4–7 | 8–12 | 13–17 | 18–19 | 20–31 |
| turbine inlet | 4.30 × 5.90 | | | | | | | |
| fish ladder | 0.60 × 0.30 | 50 | | | | | | |
| floodgate NW | 0.43 × 0.20 | 35 | | | | | | |
| floodgate NW | 0.43 × 0.30 | 35 | | | | | | |
| floodgate SE | 1.00 × 0.30 | 19 | | | | | | |
| trash gate | 0.70 × 0.40 | 0.3 | | | | | | |
| | | | | | | | | |
| river temperature (°C) | | | 8.9 | 8.9 | 9.8 | 11.1 | 13.4 | 15.6 |
| river discharge (m³ s⁻¹) | | | 4.7 | 4.5 | 4.6 | 4.4 | 4.7 | 3.7 |

The statistical software R [27] was used for data inspection and statistical analyses. Differences in descent trajectories between the RSTs were tested using a bootstrapping routine applied to the Kolmogorov–Smirnov test [28,29]. This routine allows for distribution ties [30]. The tests were run using the ks.boot-function in the matching library of R [28]. The calculations of daily fish guidance efficiency (FGE) were based on tagged smolts using the open surface gate, divided by the tagged smolts that were available for migration. Smolts that were available for migration were calculated based on released tagged smolts from the start of the study, subtracting the fish detected in the turbine tailrace and the surface gates the previous days. The length difference between turbine and surface gate migrants and the potential effect of distance from the turbine to the surface gates were tested using linear models both as linear predictors and as polynomials of degree 2 (to allow for possible minimum/maximum effects). In addition, a piecewise regression model was fitted to explore breakpoint values for the distance to the turbine intake effect on daily FGEs [31]. This piecewise regression was conducted using the segmented R library. Model selection was based on Akaike's information criterion (AIC) [32,33].

In order to quantify and compare the timing of migration between release cohorts and migration routes, candidate time-to-event models were fitted using the survival library in R [34]. As predictors, we used day of release (integer) and before/after opening trash gate (categorical, BA_TG), and migration routes were included as a group effect. Candidate models, using various additive and multiplicative combinations of these three predictors, were fitted as the Cox proportional hazards model that was subjected to model selection using AIC [35,36]. For this analysis, only individuals that were detected after release were used.

## 3. Results

Atlantic salmon smolts 2010 cohort in River Storelva were on average $139.0 \pm 14.5$ mm (s.d.) in total length, and the dominating age at smolting was 2 years. Untagged smolts were captured almost daily in RSTs along the river migration route during the 2010 smolt run, at the trap upstream the hydropower plant ($n = 4832$), in the tailrace ($n = 3487$) and in the river mouth ($n = 726$) (figure 2). The smolt run started in late April and ended at the end of May. The day with 50% cumulative smolt descent was 3 days earlier in the trap upstream the hydropower plant compared with the trap in the tailrace, and the accumulated catch trajectories at the two RSTs were significantly different (two-sample Kolmogorov–Smirnov test, $D_{\text{Fosstveit}} = 0.588$, $p < 0.001$). Furthermore, catches in the river mouth RST were 7 days later than in the tailrace RST (difference in 50% cumulative descent), and the accumulated catch trajectories at the two RSTs were significantly different (two-sample Kolmogorov–Smirnov test, $D_{\text{river mouth}} = 0.794$, $p < 0.001$).

During the smolt run, tagged smolt could move past the dam using either the turbine tunnel or one of the four surface gates. The turbine migration route was open throughout the smolt run and 451 smolts

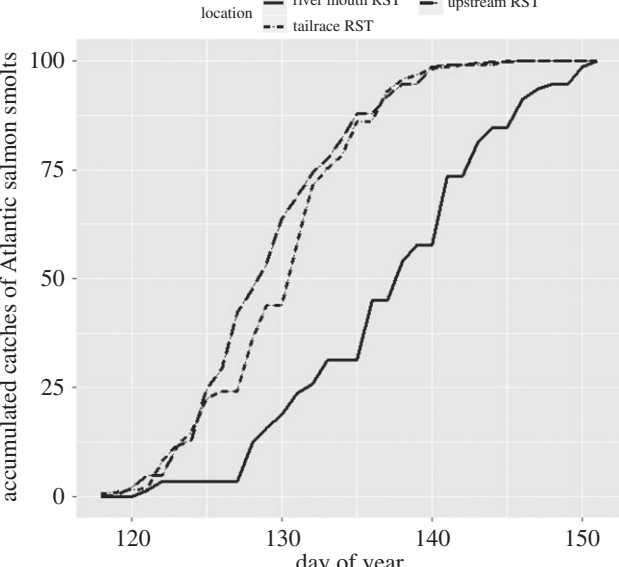

**Figure 2.** Accumulated daily catches of Atlantic salmon smolts in upstream, tailrace and river mouth RST.

used this as their migration route. A total of 231 smolts were never recaptured, while 239 used one of the different surface gates (table 2). The surface gates were opened sequentially during the smolt run, starting with the gate farthest away from the turbine. The FGE varied from 0% to 33.8%, and the highest guidance efficiency was achieved when the trash gate closest to the turbine inlet was opened. A linear model, including distance to the turbine inlet as explanatory variable, predicted the highest FGE for the gate closest to the turbine inlet (trash gate; figure 3). Trash gate migrants were significantly larger than the turbine migrants ($p < 0.0001$). Note that turbine migrants also include smolts that were detected while dead in the tailrace due to turbine blade strike ($n = 16$).

From the selected time-to-event model, it was estimated that turbine migrants (progression coefficient $2.178 \pm 0.465$) spent a shorter time in the forebay before migrating compared with the floodgate migrants (coefficient $= 0.465$) (table 3 and figure 4$a$). However, the fastest progression rate was found for trash gate migrants (5.056). Even though start day had a significant effect on migration probability, the predicted migration probability trajectories were not very different among release cohorts for the before trash gate opening migrants (figure 4). However, because route was involved in significant interactions with both before/after opening trash gate (i.e. route*BA_TG) and with start day (i.e. route*start), this resulted in a substantial cohort effect for the trash gate migrants. In particular, early release trash gate migrant cohorts had high initial migration probabilities (typically greater than 0.7) at the opening day of the trash gate but with relatively gentle response slope as time progressed (figure 4). Later release trash gate migrant cohorts had lower initial migration probabilities (approx. 0.5) that rapidly progressed to cumulated migration probability of 1. After opening the trash gate, the model predicted much higher probabilities for using the trash gate alternative than the other alternatives (figure 5).

## 4. Discussion

It is often assumed that the construction of a fish passage automatically restores functional river connectivity. In this study, 22 out of 921 tagged Atlantic salmon smolts used the floodgates and the fish ladder during the initial 20-day period when the trash gate near the turbine intake was closed. However, during this time a part of the population migrated through the turbine, while others waited in the forebay. In the last days of the smolt run, the trash gate, nearby the turbine inlet, was opened and considerable smolt migration occurred through this migration route. This demonstrates how just a few metres' misplacement of a surface bypass may substantially decrease the probability of succeeding with a fish bypass at a power plant intake. Moreover, since both delayed migration and migration through the turbine are associated with high mortality, this suggests potentially strong selection processes at hydropower plants.

Generally, national fish guidelines recommend that the bypass should be placed close to the trash rack or other guidance structures [37–39]. However, there are few case studies testing this recommendation.

**Table 2.** Number of PIT-tagged Atlantic salmon smolts released upstream of Fosstveit hydropower dam and the number of smolt migrating at the different migration routes. In addition, a calculation of accumulated smolts in the forebay each day is added (fish from previous release + daily release − migration routes). Shaded areas correspond to days when the gates were closed.

| | PIT-tagged smolts | accumulated in forebay | migration routes | | | | | |
| | | | turbine tunnel | fish ladder | floodgate NW (20 cm) | floodgate NW (30 cm) | floodgate SE | trash gate |
|---|---|---|---|---|---|---|---|---|
| 30 April | 9 | 9 | 0 | | | | | |
| 1 May | 87 | 96 | 0 | 0 | | | | |
| 2 May | 0 | 43 | 53 | 0 | | | | |
| 3 May | 127 | 165 | 5 | 0 | | | | |
| 4 May | 0 | 128 | 37 | | 0 | | | |
| 5 May | 76 | 204 | 0 | | 0 | | | |
| 6 May | 51 | 220 | 34 | | 1 | | | |
| 7 May | 39 | 254 | 5 | | 0 | | | |
| 8 May | 49 | 247 | 53 | | | 3 | | |
| 9 May | 53 | 279 | 19 | | | 2 | | |
| 10 May | 60 | 337 | 2 | | | 0 | | |
| 11 May | 54 | 358 | 32 | | | 1 | | |
| 12 May | 70 | 369 | 58 | | | 1 | | |
| 13 May | 0 | 340 | 28 | | | | 1 | |
| 14 May | 33 | 364 | 5 | | | | 4 | |
| 15 May | 31 | 393 | 1 | | | | 1 | |
| 16 May | 6 | 360 | 35 | | | | 4 | |
| 17 May | 41 | 400 | 0 | | | | 1 | |
| 18 May | 2 | 372 | 28 | 2 | | | | |
| 19 May | 36 | 389 | 18 | 1 | | | | |
| 20 May | 1 | 316 | 4 | | | | | 70 |
| 21 May | 96 | 325 | 18 | | | | | 69 |
| 22 May | 0 | 231 | 16 | | | | | 78 |
| | 921 | 231 | 451 | 3 | 1 | 7 | 11 | 217 |

In the present study, we demonstrated that how just a few metres' misplacement of a surface bypass may substantially decrease the probability of succeeding with a fish bypass at a power plant intake and further highlighted the importance of assessing passage structures and their efficiency. A short distance between water intake and bypass structure is essential, and a recent study on radio-tagged Atlantic salmon demonstrated how smolts preferred the surface gate located closest to the turbine intake when several other gates further away were available for migration [40]. Downstream migrating smolts are mainly surface orientated and follow the main river flow. In forebays, the main current velocity leads to the turbine intake. We hypothesize that smolts first start their search for other alternative migration routes when facing the dark turbine intake covered with a trash rack. If the alternative migration routes are placed too far away the smolts struggle to locate them. Thus, the findings in our study and the study performed by Havn *et al*. [40] present empirical support to the general advice that the placement of surface bypasses in relation to the turbine intake is important for the FGE, and that it should be placed close to the inlet trash rack or other guiding structures like louvre deflectors or bobble screens.

The data show an up to 20-day delay for bypass migrants due to inadequate placement of the surface bypass. The delay would probably have been even longer if the trash gate close to the intake trash rack had not been opened towards the end of the smolt run. The delayed smolt may suffer high predation

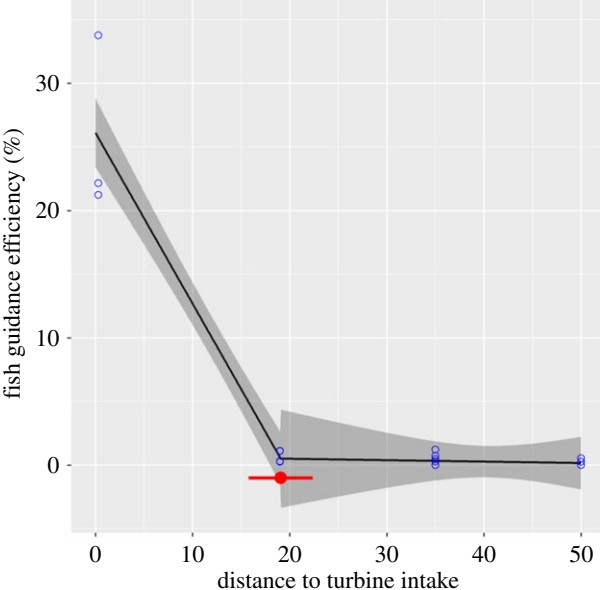

**Figure 3.** Predicted FGE for different surface gates in the hydropower dam as the distance from the turbine intake (m). Breakpoint estimate with corresponding standard error bars is shown in red. Shaded areas correspond to 95% confidence bounds.

**Table 3.** Cox proportional hazards parameter estimates for the selected time-to-event model estimation migration trajectories for salmon smolt descending the Storelva river system. BA_TG = before/after opening the trash gate (two levels). Model fit statistics: concordance = 0.771 (0.013, s.e.), $R^2$ = 0.561; likelihood ratio test: $\chi^2$ = 569.1, d.f. = 7, $p < 0.0001$.

| term | coef | exp(coef) | s.e.(coef) | Z | Pr(>\|z\|) |
|---|---|---|---|---|---|
| start | 0.465 | 1.592 | 0.076 | 6.080 | <0.0001 |
| route[Trash gate] | 5.056 | 156.989 | 0.828 | 6.105 | <0.0001 |
| route[Turbine] | 2.178 | 8.832 | 0.644 | 3.382 | 0.0007 |
| BA_TG[Before] | 6.351 | 572.924 | 0.535 | 11.877 | <0.0001 |
| start*route[Trash gate] | −0.300 | 0.741 | 0.076 | −3.916 | <0.0001 |
| start*route[Turbine] | −0.200 | 0.819 | 0.069 | −2.884 | 0.0039 |
| start*BA_TG[Before] | −0.266 | 0.766 | 0.034 | −7.764 | <0.0001 |

levels, elevated energetic costs and decreased migration speed [18–20]. Normally, the smolts enter the coastal waters at a time with optimal temperature and food supply [17]. The importance of this optimal migration window is demonstrated by smolts entering the coastal waters at other times that have low survival to adults [41]. In addition, several smolts were not recaptured after release in our study, which indicates that some smolts lost motivation upstream of the dam and stopped migrating, suffered predation or died. Alternatively, they migrated through PIT antennas and traps without being registered. This last alternative is highly unlikely due to the total detection/encounter probability through the system of PIT antennas and RSTs being close to 1 [24]. If smolts are prevented from reaching seawater, a partial re-adaptation to freshwater will occur, known as de-smolting or parr-reversion [42]. Our findings indicate that a part of the smolt population might postpone migration if only turbine migration and a misplaced bypass are available as migration routes. Much effort has been made to develop fish-friendly turbines [43], thus our findings highlight another aspect of this development. Even though the turbine is fish friendly with high survival for turbine migrating fish, there could still be characteristics at the turbine intake that will prevent a part of the smolt population from migrating. A combination of both surface bypasses and fish-friendly turbines could be a provident mitigation measure that allows safe downstream migration for smolt with individual migration preferences.

The smolts that used the turbine as a migration route were smaller than the smolts that waited and migrated through the trash gate. Potentially, the trash rack could function as a strainer, only letting

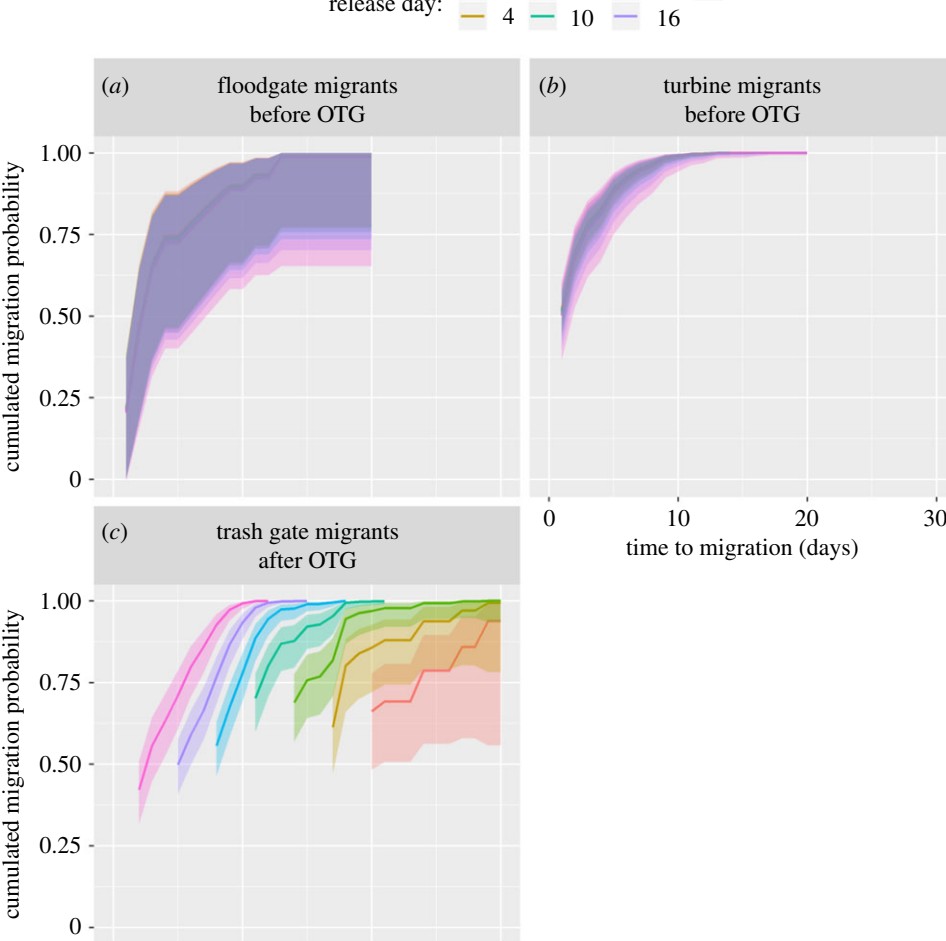

**Figure 4.** Prediction plot for the cumulated migration probabilities for the selected Cox proportional hazards model. For the floodgate (*a*) and turbine migrants (*b*), the before open trash gate (OTG) period is plotted only, as trash gate migrants do not have migration opportunities during this period. The different colours reflect different release cohorts, but they are hard to separate as the confidence bounds are largely overlapping. (*c*) The predicted trash gate migrants' cumulated migration trajectories for seven release cohorts (different colours) for the OTG period. The most recent cohorts appear from the left in the figure. Shaded areas correspond to 95% confidence bounds.

through the smallest individuals. However, the rack spacing is rather large (50 mm) and Haraldstad *et al.* [44] document that sea trout kelts (*S. trutta*) up to 450 mm migrated through this trash rack during spring descent. This lends support to the assumption that there are other mechanisms, such as behavioural traits, in addition to length that could explain smolt preference for different migration routes. It is clear that larger smolts have a greater capacity to withstand high water velocities over time compared with small fish [45]. Smolts might hold their position in front of the rack for a period and only the best swimmers (large individuals) resist the strong current. However, a study by Peake and McKinley [46] demonstrated that wild Atlantic salmon smolts of 124–211 mm in length did not show differences in swimming capacity against currents up to 1.26 m s$^{-1}$. Considering the large rack spacing and that the water velocity at the tunnel inlet area is low (less than 0.19 m s$^{-1}$) at Fosstveit HEP, then this suggests a minor contribution of size and swimming capacities to the factors underlying individual differences in migration routes in our study. Thus, contrasting behavioural profiles may be an underlying factor to the observed size differences between turbine and bypass migrating smolts in the present study.

Time spent in the forebay was fairly similar among turbine migrants and did not depend on time until the trash gate was open when the majority of the fish migrated. It is possible that this constancy in time to migrate through the turbine represents a certain behavioural profile in smoltified salmon. It

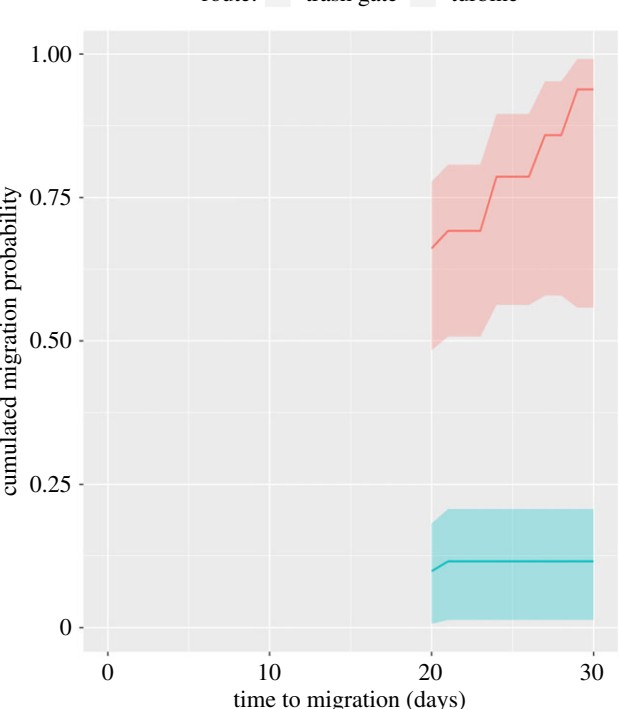

**Figure 5.** Prediction plot for the cumulated migration probabilities for the selected Cox proportional hazards model focusing on the 'after trash gate opening' period. Trash gate and turbine migrants are plotted only, as no migrants used floodgate alternatives during this period. The plotted trajectories are for individuals that were released the day before opening the trash gate. Shaded areas correspond to 95% confidence bounds.

has been documented that behavioural traits are often organized in suites of traits that show constancy across context and time, representing different behavioural syndromes within a population [47]. Moreover, such an individual variation has been associated with life-history traits [48], and in a recent review, Mittelbach *et al.* [49] pointed out that little attention has been paid to the ecological consequences of the varying behavioural phenotypes in wild populations. The results from our study point towards selection processes operating on the behavioural axis in delayed migrants versus turbine migrants. Despite the expected increase in mortality for turbine migrants due to turbine blade strikes, the surviving turbine migrants may experience higher post-smolt survival compared with smolts that experience significant delays in migration [15]. Hence, there might be complex trade-offs between acute survival costs (via turbines) for the benefit of optimal sea entrance timing versus acute survival maximization (via bypass) at the cost of suboptimal sea entrance timing. The ultimate outcome of this selection game remains to be elucidated by lifetime survival and reproduction data. Moreover, because growth rate and the behavioural profile of an individual often are linked to each other [50], contrasting behavioural profiles may be an underlying factor to the observed size differences between turbine and bypass migrating smolts in the present study. Further studies are needed to untangle the interplay between size- and behaviour-dependent selection at hydroelectric power plants and their potential population-level effects.

These results emphasize that timing and placement of mitigation measures are important for optimal management of Atlantic salmon. Moreover, it sheds light on the potential selection processes at hydropower intakes, stressing that both behaviour and size should be included as important traits under selection in wild Atlantic salmon populations in regulated rivers.

Ethics. Permission to catch Atlantic salmon smolt in River Storelva was granted by the County Governor of Aust-Agder. PIT-tagging of fish was approved by the Norwegian Animal Research Authority, NARA (Forsøksdyrutvalget, FDU, FOTS ID 2447).

Data accessibility. All data and R scripts used in the present study are available from the Dryad Digital Repository: https://dx.doi.org/10.5061/dryad.8b876q6 [51].

Authors' contributions. F.K. designed the study; F.K. and T.H. collected and prepared data for analysis; T.H., F.K., E.M.O., T.O.H. and E.H. analysed the data and interpreted the results; T.H. drafted the manuscript; T.H., F.K., E.M.O., T.O.H. and E.H. were involved in finalizing the manuscript. All authors gave final approval for publication.

Competing interests. The authors declare no competing interests.

Funding. This work is a part of T.H.'s PhD, funded by Agder Energy, Norwegian Environmental Agency and Norwegian Institute for Water Research.

Acknowledgements. We thank Kate Hawley, Åsmund Johansen, Anders Karlsson, Carolyn Rosten, Torstein Kristensen and Espen Lund who assisted with catching and tagging smolts in the field. Special thanks are given to Jim Güttrup for his indispensable contribution during years of fieldwork in River Storelva.

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
