## [Reviewer comments · Royal Society Open Science]

Review History

RSOS-181741.R0 (Original submission)

Review form: Reviewer 1 (Andrew Harbicht)

Is the manuscript scientifically sound in its present form?

No

Are the interpretations and conclusions justified by the results?

No

Is the language acceptable?

No

Is it clear how to access all supporting data?

No

Do you have any ethical concerns with this paper?

No

Have you any concerns about statistical analyses in this paper?

Yes

Recommendation?

Major revision is needed (please make suggestions in comments)

Comments to the Author(s)

Manuscript overview:

This manuscript describes a study in which downstream migrating salmon were intercepted above a dam, tagged with PIT tags, and released into the reservoir/forebay of the dam. Throughout the migration period, the migrating salmon were presented with multiple passage options: through the active turbines (all the time) or through one of several surface bypass options (alternating). The overall results indicate that salmon smolt were highly likely to use the turbines despite safer options being available. Once a trash gate near the turbine intake was opened, this became the preferred option for migrating smolt. The authors use this to justify a claim that proximity to a turbine intake, and therefore a trash rack (which should prevent larger fish from entering the turbines) significantly improves the efficiency of alternative passage options for migrating fish.

Major points:

I think there is quite a bit of potential in the data and the work that has already been done, however, there is much more potential that can still be achieved with this data by simply conducting new analyses and by presenting the data differently. Several of the graphs provide little useful information while other key pieces of information were not displayed at all. As for the analyses, I feel like the authors are only just scratching the surface of what info may be drawn from this data. I think they should strongly consider conducting a time-to-event analyses on their data as it would allow them to quantify the effects that various parameters have on the time that smolt require to pass the dam (such analyses assume that all fish will eventually pass the dam). Rather than counting the number of fish that choose the different passage options (which was not really presented to the readers) the authors can calculate how each passage option affects a fish's probability of passing the dam at any given time, while accounting for other confounding variables such as size and temperature.

I also think that if a main result is that proximity to a turbine intake/trash rack influences passage efficiency, we should have at least seen the relationship between distance to the trash rack for each alternative and its efficiency. Also, if more info is available on the velocities within the forebay are available, they should be included. If proximity to the turbine intake does play a role, I strongly suspect it is because the main flow/current through the forebay leads almost directly to the turbine. If this is the case, improved passage efficiency could have potentially been achieved at the fish ladder by reducing turbine intake volumes and increasing spillage through the ladder. If this is the case, then we could arrive at the opposite result as the authors found. Of course, I can't be sure, but it should be considered.

I think that the results of this study are/could be important for the design of fish passage features and so this is important to get out to readers, but, that being said, I think some of the claims made in the discussion may exceed what can be said based on the results as they were presented. I am particularly referring to the claims about behavioural selection on migrating smolt. This may have occurred, but I don't think the data that the authors have can make such a claim. They can certainly suggest this as a possibility, but this should be done in a more circumspect way.

Finally, I noticed quite a few issues with the grammar throughout the manuscript. I think that these authors should consider having a native English speaker work with them to sort these issues out. They are not really a major issue, rather many small issues, but before this manuscript can be considered for publication they should be addressed.

Small and medium points:

Title:

This is not a bad title, and I understand what the authors were going for, but I feel like this study did not directly test for selection processes, but rather it identified potential (and probably) selection processes. So, perhaps a better title would be something like: "Migratory passage structures at hydropower stations: potential physiological and behavioral selective forces"

Abstract:

- line 21-23: it would be better to start this sentence with something like "in an effort to counteract such effects...". Plus, you probably don't need to get right into Atlantic salmon yet, keep it more general for the moment.

- line 26 - 28: here, and in many instances throughout the article, the authors use the term upstream as a preposition, but it needs to be in relation to something. So, use "upstream from ..." rather than just "upstream...".

- line 30 - 31: this should say "During a period of ...", since you haven't mentioned a specific period of low efficiency yet.

Introduction:

-lines 43 - 54: as with the abstract, this paragraph could really be broader, encompassing at least other salmonid species with similar life cycles. This info applies to many anadromous species, not just Atlantic salmon.

- line 47 - 48: A bit more information about the types of mitigative measures employed to facilitate migration would go a long way here. In fact, I might suggest allocating an entire paragraph to the different types of passage options employed for this purpose, with a very brief description of how each works so that the readers (who may not be fish people) understand that there are truly many different options used and no general agreement about what works and what doesn't.

-lines 62 - 83: this paragraph is huge and covers several general topics. I think you should rework this one, keeping sentence length in mind and try to break it up into 2 or 3 smaller paragraphs. It looks like there is one area of discussion about smolt migration and how it is adapted to environmental cues, another section about dams causing delays and the impacts that can have on survival, and a third section on how dams can act as selection mechanisms when multiple passage options are available to migrating smolt. These are three separate points, each of which could benefit from a little more discussion. I realize word counts are sometimes hard to meet, but this should be possible here and it would improve the flow and help readers keep different ideas clear in their heads.

-lines 84 - 88: Was there no question that the authors specifically set out to test before conducting this study? I think this paragraph should include a statement of why smolt were PIT tagged and tracked as they passed a dam and what the authors expected to observe.

Methods

-lines 92 - 94: how many HEPs are there on the river Storelva? If the Fosstveit HEP is new, when was it built? Is it the last/lowest HEP in the system?

- line 97 - 98: this is the second time that the authors use the term "trash rack" but I don't believe they have ever defined what it is to the readers who may be unfamiliar with fish passage literature.

- line 99 - 101: here the authors say that there are four migrational bypass options, but in the table the north west flood gate is doubled, so is this just two halves to the same gate, or are there actually 5 options available to migrating fish?

lines 102 - 104: I get the feeling that the water velocity will be an important factor affecting where fish go when they arrive at the dam. For this reason, it would have been very nice to know what the velocity was at each migratory option throughout the course of the experiment, and not only for the turbine intake channel.

lines 105 - 106: the authors state that fish were caught in a rotary fish trap, but figure 1 shows two traps. Did the authors only tag and release fish caught in the upstream trap, or were untagged fish caught in the lower trap also transported back up? I see the answer further down, but at this point, the reader is unsure.

lines 110 - 111: Would it be possible to alter the map figure so that the locations of the two RSTs and two antennas downstream of the dam can be displayed. I imagine that as it is, they would have appeared one on top of the other in the maps current state and that is why they were not included.

Lines 111 - 112: I think it would be a good idea, before this sentence, to remind the readers that migrating smolt could move past the dam using either the turbine, or one of the 3-4 surface gates, and that which gate was available was alternated for the purpose of the experiment.

line 115 - 116: how was the catch probability of 61.5% for the tail race RST calculated?

lines 118 - 119: only fish detected at the Fosstveit dam were used to calculate forbay holding times, but how do you know when they arrived in the forebay? Were you considering that they were in the forbay once they were released? Can you be sure none of them decided to move upstream post-release?

lines 118 - 119: What do the catches/detections in the lower two RSTs and two antennas tell you about the detection probabilities of the antenna/trap at Fosstveit. Were fish detected further downstream, but not at the dam? Is this how you calculated the tailrace detection probability?

lines 120 - 124: here the authors are talking about the effects of length on path choice, but this has not come up earlier. Was this something they were interested in since the beginning? If so, it should have been explained earlier.

lines 120:124 - I think that a major piece of information is not being considered in this analysis. By comparing sizes among individuals that use separate passage alternatives, you can make a statement about whether fish with different sizes (swimming capacities, presumably) prefer different passage options. I'm assuming (without looking ahead) you will also be looking at the frequencies and contrasting this among passage options to contrast passage efficiencies, but another major piece of information that you have (albeit not as an extremely precise form) is the time that fish require to bypass a dam. I would strongly suggest considering a time-to-event

analysis the time between release and recapture below the dam as the response variable, and the fish's size as a covariate, environmental conditions (discharge, and temperature), and finally, the currently available alternative passage route as explanatory covariates. This would greatly increase your power to say something from this data.

Results:

- line 128 - 129: I feel like figure 2 is displaying different things that should probably be displayed separately. The fish caught in the upstream trap represent the intensity of the smolt migration, but fish being caught in the tailrace RST represent fish choosing the turbines as a means of passage. Shouldn't this tailrace data be displayed along with the old riverbed antenna detections instead? Also, this may be a personal preference thing, but I believe the dates on the x-axis should be month-day, and not day-month, plus use a - or a / rather than a point.

- lines 131 - 132: Many more fish were captured in the upstream RST than were tagged. How do the authors explain this? Were they trying to tag a specific number per day, or a specific percentage of that day's catch perhaps. This is not explained.

- lines 128 - 132: This sort of summary information about the run in general is quite informative to readers familiar with fish data and could be expanded upon. I would recommend having an entire paragraph in the results section that just describes the smolt run: how many were caught, how many were recaptured downstream, a population estimate if you have it, the size distribution, etc. I like that you mentioned the 50% cumulative capture frequency and I think that something like this could make a more informative graph than figure 2, or perhaps in conjunction with figure 2. Once you have the smolt migration info out of the way, you can start with the data on path choice results.

- line 135 - 138: Am I understanding this info correctly? Were these 360 smolts never detected again, or were they only not detected at the RST and antenna at the dam. If they were detected further downstream from the dam, then they didn't stay in the reservoir and simply evaded your detection methods.

- line 143 - 145: So what happened to the other fish, the remaining 40.5%. Were they never detected again, or did they move downstream once a different passage gate was opened.

- line 145 - 147: were any statistical tests performed on these time differences?

- line 148 - 149: Panel B of Figure 3 definitely contains some telling information, but it is difficult to interpret with the way that the authors have chosen to display the information. Were they expecting a relationship between the date of release and how long fish would require to bypass the dam? If so, this was not mentioned earlier. If not, then release date should not be the x-axis of these two graphs. I would find it much more useful to see a single bar plot with passage route as the x-axis and time in the forebay as the y-axis. Then I could clearly see where the mean/medians are and I could compare the separate alternatives, something which is impossible with the graph in its current state. Another issue is that which passage options were available at which times is only available to the reader in Table 1. At a minimum this info should be included in Figure 3 if the authors wish to keep this graph as is. Finally, we can see how many fish pass the dam on a given day and (if info about which gates were open is added to the graph, we could sort of tell how that option impacts passage efficiency, but we cannot tell if the fish passing through the NW gate, for example, were in the forebay for a while already waiting for the right passage option to become available.

- line 152 - 153: this is a major result of the study in its current state, and yet it only gets one line of text and no figure? Mean size and passage option used should definitely be a figure included in this manuscript.

Discussion:

- lines 158 - 164: I think that a summary paragraph is an excellent way to begin a discussion section. The authors start of this way, but only really lightly touch on some of their results and omit other potentially important aspects before getting into implications of their work. I'd like to see this section expanded a bit to cover more of the findings and only briefly mention the implications. The next paragraph should deal more with those and how it fits into the broader scope of the available research.

- line 178 - 180: Here the authors are saying that some smolt were never detected again following release, but earlier they mentioned 360 smolt that were not detected in either the tailrace or the old riverbed. Are these the fish they are referring to, or were some of these fish detected further down and this was just not mentioned earlier on.

- lines 193 - 196: I think that this option is being dismissed a little too easily. Yes, it may be possible for larger fish to get through the trash rack, but there may be behavioral traits that prevent larger salmon from trying to squeeze through the trash rack. Traits which differ between salmon and trout. Trash rack spacing can have an effect on behaviour prior to bar spacing physically limiting passage.

- line 198 - 199: is this statement clear based on the data you've provided? I would argue that it is not as you have not provided us with much data on water velocities throughout the forebay. Maybe you are referring to previous research, but in that case it needs to be cited.

-line 215 - 221: While I do not believe you have actually displayed solid evidence of behavioural differences in the tagged fish, I do believe that some likely exist and they could influence willingness to enter the turbine intake tunnel, so this is an interesting point. However, I think that throughout the discussion, you will need to tone down the argument that behavioural selection has been detected. Instead, I would suggest discussing the potential for behavioural selection to occur in such a situation.

Review form: Reviewer 2

Is the manuscript scientifically sound in its present form?

No

Are the interpretations and conclusions justified by the results?

No

Is the language acceptable?

Yes

Is it clear how to access all supporting data?

No

Do you have any ethical concerns with this paper?

No

Have you any concerns about statistical analyses in this paper?

Yes

Recommendation?

Reject

Comments to the Author(s)

This study assessed differences in downstream migration of Atlantic Salmon smolts across different bypass routes at a hydroelectric facility. The authors used a capture-mark-recapture design to determine routes used by tagged smolts. I think the idea of assessing selection at fish passage facilities is a very interesting one, and downstream passage efficiency in general is understudied for many species and facilities. Many fish passage studies provide evidence for differential passage-success among individuals, but I do not think the consequences of selection have been fully appreciated. That being said, I think the manuscript needs attention to clarify questions being asked and how the data are used to answer them. I found it difficult to follow the accounting of tagged fish because the description of PIT antenna and trap placements is lacking in detail. Additionally, the experimental design is not introduced very clearly in the Methods section. It appears from Table 1, the authors have designed an experiment to test how placement of a bypass route from a turbine inlet influences migration delay (regression design). However, the analysis and inference does not match with that design. As far as I can tell, the only formal analysis occurred when assessing size differences among bypass routes. I think the authors could do a better job of clearly stating their questions to be answered, and this would help structure the Results and analysis around answering those questions. The experimental design appears to be such that authors are testing a “distance” effect, but there is no formal analysis of this. I think this is the biggest flaw of the manuscript. Finally, I think the Discussion would be much improved if the authors would more fully consider and discuss potential mechanisms for the “distance” effect their data shows. I provided more specific line-by-line comments below.

L78: It is not clear what the authors are referring to as behavior characters and life history traits. Swimming capacity is more of a physiological constraint rather than a consequence of behavior. Re-word this to either include another behavior character example or better yet mentioning a life history trait example and a behavior character example. This would transition into the next sentence more smoothly.

L81-82: The mechanism is mortality, but what we lack is knowledge on the consequences of selection.

L84: This paragraph could be strengthened by providing readers with predictions made regarding migration options. The authors appear to have some thoughts on the quality of migration options (i.e., distance from turbine channel). Clearly and explicitly stating the questions asked in this study would help guide the reader through the analyses and improve the clarity of the manuscript.

L99-101: Are there differences in heights of the different bypass routes? If a fish goes over a floodgate, is it falling 14.5 m? Could the threat of falling that far play into choices of bypass route use?

L102: Please clarify if this “water velocity at the tunnel inlet” is in reference to the turbine tunnel. Also, do you know the velocities and/or discharges through the different bypass routes? That could provide some information on why fish are using different routes, or what cues they are using to decide on.

L107: Please include more detail on tagging protocol. Were tags injected via hypodermic needle or placed inside fish via incision?

L109: This is fine to mention the total number of tagged fish, but here would be a good place to provide more detail on your experimental design. It appears from Table 1 that you have a design to test the effect of distance to the turbine inlet of varying bypass routes, but this isn't talked

about in enough detail in the Methods. Why is the fish ladder opened up again during 18-19 May?

L110: You say four PIT antennas here, but below only describe locations for three. What are the details on the fourth antenna?

L117: It is unclear what these percentages are representing. Please clarify.

L118: The authors need to describe their antenna array more clearly. It is difficult to know which antennas are used to assess fish use of different migration routes. On line 116, it reads as if both turbine and surface gate migrants can be detected by the same antenna. If that's the case, how are authors able to differentiate migration routes?

Did PIT antennas cover the entire width of the river? What about RSTs?

L120: Please include more detail about your statistical analysis here. For instance, did you check assumptions of ANOVA? Your supplementary code does not show any formal statistical test of assumptions or coding for plotting residual patterns. Also, what critical value are you using?

L122: Please name the specific post-hoc test you used.

General Results comment:

The way the results are presented relies a lot on presenting migration route use prior to 20 May and post 20 May, which does not take advantage of their experimental design. Their design is set up to test a "distance" effect as the bypass route is systematically moved closer to the turbine inlet channel over time (with an exception of the fish ladder being reopened late in the study). It's not clear why the authors are not analyzing and presenting their data in that context.

L129-130: What information is this data providing in regards to your objectives in this study? I think stating clear questions upfront in the Introduction would help organize the Results.

I could see how assessing the differences in the number of fish captured in upstream and downstream traps could be meaningful, but these numbers aren't put into context with the objectives. Are these numbers directly comparable? Are there differences in capture probability between these two traps? What's the significance of the 50% cumulative abundance being different by 3 days, and could this be an artifact of differences in capture probabilities? What day was this?

L132: I think presenting data from antenna detections and recaptures separately would be informative rather than combining them across methods.

L139: So now, your pool of potential re-encounters is 360+97 based on hold-outs from prior to 20 May and the newly tagged fish? You should clarify this to put your numbers below in context. But again, I'm not sure why data is being grouped into pre 20 May and post 20 May rather than analyzing data in the context of your design?

L142: How many? It would be good to remind the reader.

L143: Where are the other ~40%?

L145: Please clarify what this value in parentheses represents.

L148: Here and elsewhere, if you're going to place emphasis on differences, then I think you should conduct a formal test to assess whether a statistical difference exists or not. The only place you do this is for the length data.

L148: If all these values in parentheses are means plus/minus SD, then inform the reader of that at first mention (i.e., L145). Then, be consistent throughout. Also, be consistent with abbreviations. Either spell out days every time or abbreviate it.

L150: Report the actual value instead saying a "large fraction".

L153: Please include more details on your statistics. Is the p-value from the post-hoc test? What's the F-value, degrees of freedom, etc.?

L158-159: The experimental design is set up to test a "distance" effect. Not clear how opening the trash gate would influence use of other migration routes? The data is pretty clear (Fig 3) that the closer the bypass is to the turbine channel the less time fish spend in the forebay.

L167: Why do you think you saw a "distance" effect? What is it about the turbine channel area that is different from the other migration routes?

L181: The validity of this assumption about detection probabilities is difficult for the reader to evaluate. Updating Figure 1 with precise locations of antenna and RSTs would help to evaluate

the likelihood of not re-encountering fish. For example, the detection probability of one PIT antenna is reported as 1, but there are three other antennas.

L204: What about behavioral avoidance as a function of size? Larger individuals could avoid the trash rack spacings more so than small individuals, despite both being physically capable of moving through the spacings.

L206: This was not tested.

Figure 1: This figure needs more detail on antenna and RST locations. Four antennas were mentioned in the Methods section, but are represented by only a single arrow. The caption is also unclear because the figure lacks specific locations of recapture/redetection locations.

Figure 2: Please format the x-axis to match date formats presented elsewhere in the manuscript.

Figure 3: The fish ladder gets re-opened on 18-19 May, but the data still show a lowered forebay time on those days. That is in conflict with the general distance effect shown by the rest of the data. Why might that be? It's also not clear why you're not analyzing and presenting this data in-line with your experimental design (i.e., x-axis would be distance from turbine channel, analysis would be based on regression).

Figure 4: Please include the number of individuals for each category. I think you should also include your statistical result with the figure. As it is, it requires a lot of details from the text to interpret this figure. Please clarify, "including fish that were not recaptured". I assumed this length data was "length at capture".

Table 1: Why does the fish ladder get reopened on 18-19 May? What is that testing?

Table 2: How did fish go through the trash gate prior to 20 May if it was not open?

Supplementary material not cited in text.

Decision letter (RSOS-181741.R0)

13-Nov-2018

Dear Mr Haraldstad:

Manuscript ID RSOS-181741 entitled "Selection processes at hydropower turbine intakes shaped by location of bypass routes during smolt migration in Atlantic salmon (*Salmo salar*)" which you submitted to Royal Society Open Science, has been reviewed. The comments from reviewers are included at the bottom of this letter.

In view of the criticisms of the reviewers, the manuscript has been rejected in its current form. However, a new manuscript may be submitted which takes into consideration these comments.

Please note that resubmitting your manuscript does not guarantee eventual acceptance, and that your resubmission will be subject to peer review before a decision is made.

Your resubmitted manuscript should be submitted by 13-May-2019. If you are unable to submit by this date please contact the Editorial Office.

Please note that Royal Society Open Science will introduce article processing charges for all new submissions received from 1 January 2018. Charges will also apply to papers transferred to Royal Society Open Science from other Royal Society Publishing journals, as well as papers submitted as part of our collaboration with the Royal Society of Chemistry (<http://rsos.royalsocietypublishing.org/chemistry>). If your manuscript is submitted and accepted for publication after 1 Jan 2018, you will be asked to pay the article processing charge, unless you request a waiver and this is approved by Royal Society Publishing. You can find out more about the charges at <http://rsos.royalsocietypublishing.org/page/charges>. Should you have any queries, please contact openscience@royalsociety.org.

on behalf of Dr Michael Tobler (Associate Editor) and Professor Kevin Padian (Subject Editor)
openscience@royalsociety.org

Associate Editor Comments to Author (Dr Michael Tobler):

We have received the feedback from two reviewers that see a lot of merit in the study. However, they also identified significant short-comings associated with data analysis and presentation, requiring substantial revisions before the manuscript can be accepted for publication.

Reviewers' Comments to Author:

Reviewer: 1

Comments to the Author(s)

Manuscript overview:

This manuscript describes a study in which downstream migrating salmon were intercepted above a dam, tagged with PIT tags, and released into the reservoir/forebay of the dam. Throughout the migration period, the migrating salmon were presented with multiple passage options: through the active turbines (all the time) or through one of several surface bypass options (alternating). The overall results indicate that salmon smolt were highly likely to use the turbines despite safer options being available. Once a trash gate near the turbine intake was opened, this became the preferred option for migrating smolt. The authors use this to justify a claim that proximity to a turbine intake, and therefore a trash rack (which should prevent larger fish from entering the turbines) significantly improves the efficiency of alternative passage options for migrating fish.

Major points:

I think there is quite a bit of potential in the data and the work that has already been done, however, there is much more potential that can still be achieved with this data by simply conducting new analyses and by presenting the data differently. Several of the graphs provide little useful information while other key pieces of information were not displayed at all. As for the analyses, I feel like the authors are only just scratching the surface of what info may be drawn from this data. I think they should strongly consider conducting a time-to-event analyses on their data as it would allow them to quantify the effects that various parameters have on the time that

smolt require to pass the dam (such analyses assume that all fish will eventually pass the dam). Rather than counting the number of fish that choose the different passage options (which was not really presented to the readers) the authors can calculate how each passage option affects a fish's probability of passing the dam at any given time, while accounting for other confounding variables such as size and temperature.

I also think that if a main result is that proximity to a turbine intake/trash rack influences passage efficiency, we should have at least seen the relationship between distance to the trash rack for each alternative and its efficiency. Also, if more info is available on the velocities within the forebay are available, they should be included. If proximity to the turbine intake does play a role, I strongly suspect it is because the main flow/current through the forebay leads almost directly to the turbine. If this is the case, improved passage efficiency could have potentially been achieved at the fish ladder by reducing turbine intake volumes and increasing spillage through the ladder. If this is the case, then we could arrive at the opposite result as the authors found. Of course, I can't be sure, but it should be considered.

I think that the results of this study are/could be important for the design of fish passage features and so this is important to get out to readers, but, that being said, I think some of the claims made in the discussion may exceed what can be said based on the results as they were presented. I am particularly referring to the claims about behavioural selection on migrating smolt. This may have occurred, but I don't think the data that the authors have can make such a claim. They can certainly suggest this as a possibility, but this should be done in a more circumspect way.

Finally, I noticed quite a few issues with the grammar throughout the manuscript. I think that these authors should consider having a native English speaker work with them to sort these issues out. They are not really a major issue, rather many small issues, but before this manuscript can be considered for publication they should be addressed.

Small and medium points:

Title:

This is not a bad title, and I understand what the authors were going for, but I feel like this study did not directly test for selection processes, but rather it identified potential (and probably) selection processes. So, perhaps a better title would be something like: "Migratory passage structures at hydropower stations: potential physiological and behavioral selective forces"

Abstract:

- line 21-23: it would be better to start this sentence with something like "in an effort to counteract such effects...". Plus, you probably don't need to get right into Atlantic salmon yet, keep it more general for the moment.

- line 26 - 28: here, and in many instances throughout the article, the authors use the term upstream as a preposition, but it needs to be in relation to something. So, use "upstream from ..." rather than just "upstream...".

- line 30 - 31: this should say "During a period of ...", since you haven't mentioned a specific period of low efficiency yet.

Introduction:

-lines 43 - 54: as with the abstract, this paragraph could really be broader, encompassing at least other salmonid species with similar life cycles. This info applies to many anadromous species, not just Atlantic salmon.

- line 47 - 48: A bit more information about the types of mitigative measures employed to facilitate migration would go a long way here. In fact, I might suggest allocating an entire paragraph to the different types of passage options employed for this purpose, with a very brief description of how each works so that the readers (who may not be fish people) understand that there are truly many different options used and no general agreement about what works and what doesn't.

-lines 62 - 83: this paragraph is huge and covers several general topics. I think you should rework this one, keeping sentence length in mind and try to break it up into 2 or 3 smaller paragraphs. It looks like there is one area of discussion about smolt migration and how it is adapted to environmental cues, another section about dams causing delays and the impacts that can have on survival, and a third section on how dams can act as selection mechanisms when multiple passage options are available to migrating smolt. These are three separate points, each of which could benefit from a little more discussion. I realize word counts are sometimes hard to meet, but this should be possible here and it would improve the flow and help readers keep different ideas clear in their heads.

-lines 84 - 88: Was there no question that the authors specifically set out to test before conducting this study? I think this paragraph should include a statement of why smolt were PIT tagged and tracked as they passed a dam and what the authors expected to observe.

Methods

-lines 92 - 94: how many HEPs are there on the river Storelva? If the Fosstveit HEP is new, when was it built? Is it the last/lowest HEP in the system?

- line 97 - 98: this is the second time that the authors use the term "trash rack" but I don't believe they have ever defined what it is to the readers who may be unfamiliar with fish passage literature.

- line 99 - 101: here the authors say that there are four migrational bypass options, but in the table the north west flood gate is doubled, so is this just two halves to the same gate, or are there actually 5 options available to migrating fish?

lines 102 - 104: I get the feeling that the water velocity will be an important factor affecting where fish go when they arrive at the dam. For this reason, it would have been very nice to know what the velocity was at each migratory option throughout the course of the experiment, and not only for the turbine intake channel.

lines 105 - 106: the authors state that fish were caught in a rotary fish trap, but figure 1 shows two traps. Did the authors only tag and release fish caught in the upstream trap, or were untagged fish caught in the lower trap also transported back up? I see the answer further down, but at this point, the reader is unsure.

lines 110 - 111: Would it be possible to alter the map figure so that the locations of the two RSTs and two antennas downstream of the dam can be displayed. I imagine that as it is, they would have appeared one on top of the other in the maps current state and that is why they were not included.

Lines 111 - 112: I think it would be a good idea, before this sentence, to remind the readers that migrating smolt could move past the dam using either the turbine, or one of the 3-4 surface gates, and that which gate was available was alternated for the purpose of the experiment.

line 115 - 116: how was the catch probability of 61.5% for the tail race RST calculated?

lines 118 - 119: only fish detected at the Fossveit dam were used to calculate forbay holding times, but how do you know when they arrived in the forebay? Were you considering that they were in the forbay once they were released? Can you be sure none of them decided to move upstream post-release?

lines 118 - 119: What do the catches/detections in the lower two RSTs and two antennas tell you about the detection probabilities of the antenna/trap at Fossveit. Were fish detected further downstream, but not at the dam? Is this how you calculated the tailrace detection probability?

lines 120 - 124: here the authors are talking about the effects of length on path choice, but this has not come up earlier. Was this something they were interested in since the beginning? If so, it should have been explained earlier.

lines 120:124 - I think that a major piece of information is not being considered in this analysis. By comparing sizes among individuals that use separate passage alternatives, you can make a statement about whether fish with different sizes (swimming capacities, presumably) prefer different passage options. I'm assuming (without looking ahead) you will also be looking at the frequencies and contrasting this among passage options to contrast passage efficiencies, but another major piece of information that you have (albeit not as an extremely precise form) is the time that fish require to bypass a dam. I would strongly suggest considering a time-to-event analysis the time between release and recapture below the dam as the response variable, and the fish's size as a covariate, environmental conditions (discharge, and temperature), and finally, the currently available alternative passage route as explanatory covariates. This would greatly increase your power to say something from this data.

Results:

- line 128 - 129: I feel like figure 2 is displaying different things that should probably be displayed separately. The fish caught in the upstream trap represent the intensity of the smolt migration, but fish being caught in the tailrace RST represent fish choosing the turbines as a means of passage. Shouldn't this tailrace data be displayed along with the old riverbed antenna detections instead? Also, this may be a personal preference thing, but I believe the dates on the x-axis should be month-day, and not day-month, plus use a - or a / rather than a point.

- lines 131 - 132: Many more fish were captured in the upstream RST than were tagged. How do the authors explain this? Were they trying to tag a specific number per day, or a specific percentage of that day's catch perhaps. This is not explained.

- lines 128 - 132: This sort of summary information about the run in general is quite informative to readers familiar with fish data and could be expanded upon. I would recommend having an entire paragraph in the results section that just describes the smolt run: how many were caught, how many were recaptured downstream, a population estimate if you have it, the size distribution, etc. I like that you mentioned the 50% cumulative capture frequency and I think that something like this could make a more informative graph than figure 2, or perhaps in conjunction with figure 2. Once you have the smolt migration info out of the way, you can start with the data on path choice results.

- line 135 - 138: Am I understanding this info correctly? Were these 360 smolts never detected again, or were they only not detected at the RST and antenna at the dam. If they were detected further downstream from the dam, then they didn't stay in the reservoir and simply evaded your detection methods.

- line 143 - 145: So what happened to the other fish, the remaining 40.5%. Were they never detected again, or did they move downstream once a different passage gate was opened.

- line 145 - 147: were any statistical tests performed on these time differences?

- line 148 - 149: Panel B of Figure 3 definitely contains some telling information, but it is difficult to interpret with the way that the authors have chosen to display the information. Were they expecting a relationship between the date of release and how long fish would require to bypass the dam? If so, this was not mentioned earlier. If not, then release date should not be the x-axis of these two graphs. I would find it much more useful to see a single bar plot with passage route as the x-axis and time in the forebay as the y-axis. Then I could clearly see where the mean/medians are and I could compare the separate alternatives, something which is impossible with the graph in its current state. Another issue is that which passage options were available at which times is only available to the reader in Table 1. At a minimum this info should be included in Figure 3 if the authors wish to keep this graph as is. Finally, we can see how many fish pass the dam on a given day and (if info about which gates were open is added to the graph, we could sort of tell how that option impacts passage efficiency, but we cannot tell if the fish passing through the NW gate, for example, were in the forebay for a while already waiting for the right passage option to become available.

- line 152 - 153: this is a major result of the study in its current state, and yet it only gets one line of text and no figure? Mean size and passage option used should definitely be a figure included in this manuscript.

Discussion:

- lines 158 - 164: I think that a summary paragraph is an excellent way to begin a discussion section. The authors start off this way, but only really lightly touch on some of their results and omit other potentially important aspects before getting into implications of their work. I'd like to see this section expanded a bit to cover more of the findings and only briefly mention the implications. The next paragraph should deal more with those and how it fits into the broader scope of the available research.

- line 178 - 180: Here the authors are saying that some smolt were never detected again following release, but earlier they mentioned 360 smolt that were not detected in either the tailrace or the old riverbed. Are these the fish they are referring to, or were some of these fish detected further down and this was just not mentioned earlier on.

- lines 193 - 196: I think that this option is being dismissed a little too easily. Yes, it may be possible for larger fish to get through the trash rack, but there may be behavioral traits that prevent larger salmon from trying to squeeze through the trash rack. Traits which differ between salmon and trout. Trash rack spacing can have an effect on behaviour prior to bar spacing physically limiting passage.

- line 198 - 199: is this statement clear based on the data you've provided? I would argue that it is not as you have not provided us with much data on water velocities throughout the forebay. Maybe you are referring to previous research, but in that case it needs to be cited.

-line 215 - 221: While I do not believe you have actually displayed solid evidence of behavioural differences in the tagged fish, I do believe that some likely exist and they could influence willingness to enter the turbine intake tunnel, so this is an interesting point. However, I think that throughout the discussion, you will need to tone down the argument that behavioural selection

has been detected. Instead, I would suggest discussing the potential for behavioural selection to occur in such a situation.

Reviewer: 2

Comments to the Author(s)

This study assessed differences in downstream migration of Atlantic Salmon smolts across different bypass routes at a hydroelectric facility. The authors used a capture-mark-recapture design to determine routes used by tagged smolts. I think the idea of assessing selection at fish passage facilities is a very interesting one, and downstream passage efficiency in general is understudied for many species and facilities. Many fish passage studies provide evidence for differential passage-success among individuals, but I do not think the consequences of selection have been fully appreciated. That being said, I think the manuscript needs attention to clarify questions being asked and how the data are used to answer them. I found it difficult to follow the accounting of tagged fish because the description of PIT antenna and trap placements is lacking in detail. Additionally, the experimental design is not introduced very clearly in the Methods section. It appears from Table 1, the authors have designed an experiment to test how placement of a bypass route from a turbine inlet influences migration delay (regression design). However, the analysis and inference does not match with that design. As far as I can tell, the only formal analysis occurred when assessing size differences among bypass routes. I think the authors could do a better job of clearly stating their questions to be answered, and this would help structure the Results and analysis around answering those questions. The experimental design appears to be such that authors are testing a “distance” effect, but there is no formal analysis of this. I think this is the biggest flaw of the manuscript. Finally, I think the Discussion would be much improved if the authors would more fully consider and discuss potential mechanisms for the “distance” effect their data shows. I provided more specific line-by-line comments below.

L78: It is not clear what the authors are referring to as behavior characters and life history traits. Swimming capacity is more of a physiological constraint rather than a consequence of behavior. Re-word this to either include another behavior character example or better yet mentioning a life history trait example and a behavior character example. This would transition into the next sentence more smoothly.

L81-82: The mechanism is mortality, but what we lack is knowledge on the consequences of selection.

L84: This paragraph could be strengthened by providing readers with predictions made regarding migration options. The authors appear to have some thoughts on the quality of migration options (i.e., distance from turbine channel). Clearly and explicitly stating the questions asked in this study would help guide the reader through the analyses and improve the clarity of the manuscript.

L99-101: Are there differences in heights of the different bypass routes? If a fish goes over a floodgate, is it falling 14.5 m? Could the threat of falling that far play into choices of bypass route use?

L102: Please clarify if this “water velocity at the tunnel inlet” is in reference to the turbine tunnel. Also, do you know the velocities and/or discharges through the different bypass routes? That could provide some information on why fish are using different routes, or what cues they are using to decide on.

L107: Please include more detail on tagging protocol. Were tags injected via hypodermic needle or placed inside fish via incision?

L109: This is fine to mention the total number of tagged fish, but here would be a good place to provide more detail on your experimental design. It appears from Table 1 that you have a design to test the effect of distance to the turbine inlet of varying bypass routes, but this isn't talked

about in enough detail in the Methods. Why is the fish ladder opened up again during 18-19 May?

L110: You say four PIT antennas here, but below only describe locations for three. What are the details on the fourth antenna?

L117: It is unclear what these percentages are representing. Please clarify.

L118: The authors need to describe their antenna array more clearly. It is difficult to know which antennas are used to assess fish use of different migration routes. On line 116, it reads as if both turbine and surface gate migrants can be detected by the same antenna. If that's the case, how are authors able to differentiate migration routes?

Did PIT antennas cover the entire width of the river? What about RSTs?

L120: Please include more detail about your statistical analysis here. For instance, did you check assumptions of ANOVA? Your supplementary code does not show any formal statistical test of assumptions or coding for plotting residual patterns. Also, what critical value are you using?

L122: Please name the specific post-hoc test you used.

General Results comment:

The way the results are presented relies a lot on presenting migration route use prior to 20 May and post 20 May, which does not take advantage of their experimental design. Their design is set up to test a "distance" effect as the bypass route is systematically moved closer to the turbine inlet channel over time (with an exception of the fish ladder being reopened late in the study). It's not clear why the authors are not analyzing and presenting their data in that context.

L129-130: What information is this data providing in regards to your objectives in this study? I think stating clear questions upfront in the Introduction would help organize the Results.

I could see how assessing the differences in the number of fish captured in upstream and downstream traps could be meaningful, but these numbers aren't put into context with the objectives. Are these numbers directly comparable? Are there differences in capture probability between these two traps? What's the significance of the 50% cumulative abundance being different by 3 days, and could this be an artifact of differences in capture probabilities? What day was this?

L132: I think presenting data from antenna detections and recaptures separately would be informative rather than combining them across methods.

L139: So now, your pool of potential re-encounters is 360+97 based on hold-outs from prior to 20 May and the newly tagged fish? You should clarify this to put your numbers below in context. But again, I'm not sure why data is being grouped into pre 20 May and post 20 May rather than analyzing data in the context of your design?

L142: How many? It would be good to remind the reader.

L143: Where are the other ~40%?

L145: Please clarify what this value in parentheses represents.

L148: Here and elsewhere, if you're going to place emphasis on differences, then I think you should conduct a formal test to assess whether a statistical difference exists or not. The only place you do this is for the length data.

L148: If all these values in parentheses are means plus/minus SD, then inform the reader of that at first mention (i.e., L145). Then, be consistent throughout. Also, be consistent with abbreviations. Either spell out days every time or abbreviate it.

L150: Report the actual value instead saying a "large fraction".

L153: Please include more details on your statistics. Is the p-value from the post-hoc test? What's the F-value, degrees of freedom, etc.?

L158-159: The experimental design is set up to test a "distance" effect. Not clear how opening the trash gate would influence use of other migration routes? The data is pretty clear (Fig 3) that the closer the bypass is to the turbine channel the less time fish spend in the forebay.

L167: Why do you think you saw a "distance" effect? What is it about the turbine channel area that is different from the other migration routes?

L181: The validity of this assumption about detection probabilities is difficult for the reader to evaluate. Updating Figure 1 with precise locations of antenna and RSTs would help to evaluate

the likelihood of not re-encountering fish. For example, the detection probability of one PIT antenna is reported as 1, but there are three other antennas.

L204: What about behavioral avoidance as a function of size? Larger individuals could avoid the trash rack spacings more so than small individuals, despite both being physically capable of moving through the spacings.

L206: This was not tested.

Figure 1: This figure needs more detail on antenna and RST locations. Four antennas were mentioned in the Methods section, but are represented by only a single arrow. The caption is also unclear because the figure lacks specific locations of recapture/redetection locations.

Figure 2: Please format the x-axis to match date formats presented elsewhere in the manuscript.

Figure 3: The fish ladder gets re-opened on 18-19 May, but the data still show a lowered forebay time on those days. That is in conflict with the general distance effect shown by the rest of the data. Why might that be? It's also not clear why you're not analyzing and presenting this data in-line with your experimental design (i.e., x-axis would be distance from turbine channel, analysis would be based on regression).

Figure 4: Please include the number of individuals for each category. I think you should also include your statistical result with the figure. As it is, it requires a lot of details from the text to interpret this figure. Please clarify, "including fish that were not recaptured". I assumed this length data was "length at capture".

Table 1: Why does the fish ladder get reopened on 18-19 May? What is that testing?

Table 2: How did fish go through the trash gate prior to 20 May if it was not open?

Supplementary material not cited in text.

Author's Response to Decision Letter for (RSOS-181741.R0)

See Appendix A.

RSOS-190989.R0

Review form: Reviewer 2

Is the manuscript scientifically sound in its present form?

Yes

Are the interpretations and conclusions justified by the results?

Yes

Is the language acceptable?

No

Is it clear how to access all supporting data?

Yes

Do you have any ethical concerns with this paper?

No

Have you any concerns about statistical analyses in this paper?

No

Recommendation?

Accept with minor revision (please list in comments)

Comments to the Author(s)

Review of RSOS-190989: Migratory passage structures at hydropower plants as potential physiological and behavioral selective agents.

The revisions the authors have incorporated have improved the manuscript greatly. The new analyses are clear and the figures tell the story really well. I have some additional minor comments for the authors to consider, and overall I thought the Discussion was very solid. I did not spend much time on editorial-style comments because, as I pointed out in my last review, I think the manuscript needs the attention of an English language service to fix numerous sentence structure issues.

Abstract L18: Provide a specific number of increase here. How much did the proportion increase?

Introduction P5, L34: I don't think this paragraph is necessary. These points are good ones, but could be incorporated elsewhere with citations. Authors generally say "the river", but are they referring to rivers in general or a specific river?

Methods P7, L35: Provide a citation for these previous findings of incision healing.

Methods P7, L53: What concession? Please clarify for readers unfamiliar with this system.

Methods P8, L5: Figure 1 only depicts two traps. Put an arrow to the "rivermouth" trap in the top panel.

Methods P8, L33: Not clear what "recapture" is referring to here. If these are antenna detections then this needs clarified. When referring to physical captures, recapture would be appropriate, and for PIT antenna detections, detections would be appropriate. If the authors combine this two types of data then something like "reencounters" might be more accurate. Also, report a total sample size for this detection probability. Would it be N=250 individuals?

Methods P9, L3: divide by the total number of tagged smolts?

Results P9, L46: Report plus/minus 14.5 directly after "139.0 mm" and just put "SD" in parentheses.

Results P10, L50: Difficult to interpret what "not very different" means? It would be helpful to present numbers here so readers can assess quantitative differences.

Results P10, L51: Need to explain what this interaction term means. The letters mean nothing to readers because this term isn't explained until Table 3.

Results P10, L59: I suggest the authors provide some interpretation of this result and the difference between early- and late-release cohorts. This is likely due to late-release fish still searching for a downstream route, while early-release fish have been in the forebay for some time.

Discussion P11, L18: Something really striking to me was how little fish used the fish passage for downstream migration. I think the authors should make a bigger point of this and call attention here in the Discussion. This could be done in a sentence or two. It's often assumed that construction of a fish passage automatically restores functional connectivity, few studies assess fish passage structures, and fewer yet assess downstream migration through fish passages. The data presented in this study are pretty striking that fish passages far away from turbine channels, where I presume many are often built on hydro-power facilities, are completely ineffective for downstream migration at least for this life stage and species. This is really good information for managers to have.

Discussion P11, L36: could be occurring.

Discussion P12, L30: fish could also have died.

Discussion P12, L52: So, a behavioral barrier might exist?

Discussion P13, L3-37: This is a good paragraph.

Figure 4 and 5: Report units for “Time to migration”.

Table 3: This is a minor formatting issue, but the “Term” column appears to be vertically justified to the top of the cell, while the statistical parameters are bottom justified. This makes it difficult to quickly skim across rows. Please change.

Review form: Reviewer 3

Is the manuscript scientifically sound in its present form?

Yes

Are the interpretations and conclusions justified by the results?

Yes

Is the language acceptable?

Yes

Is it clear how to access all supporting data?

Not Applicable

Do you have any ethical concerns with this paper?

No

Have you any concerns about statistical analyses in this paper?

I do not feel qualified to assess the statistics

Recommendation?

Major revision is needed (please make suggestions in comments)

Comments to the Author(s)

Manuscript content summary:

The authors present a study on downstream migration routes of atlantic salmon at hydropower plant (HPP) Fosstveit in Norway. Within 23 days (April 30 – May 21, 2010) a total of 923 salmon smolts were caught, PIT tagged, and released upstream of the HPP. Four possible downstream migration routes exist (1) the turbine, (2) floodgates, (3) a fish ladder, and (4) a trash gate. The turbine outlet was individually monitored with a rotary screw trap equipped with nets. The remaining 3 migration routes were monitored conjointly with PIT-antennas in the residual flow reach downstream of the HPP. On May 20 and 21 the trash gate was opened, leading to altered flow conditions in the HPP forebay, such that the number of fish detected in the residual flow reach significantly increased on that 2 days.

General comments:

The manuscript is well written and the level of English is good. I consider the manuscript contents as very interesting and valuable for the research community given the below corrections and clarifications will be included.

To me, some important details such as the specific locations of the monitoring devices, the characteristics (total river discharge, turbine discharge, fish ladder discharge, floodgate discharge, floodgate dimensions, trash gate discharge, trash gate opening dimensions, ...) of all individual HPP elements and the exact numbers of detected fish are unclear and should be explained in greater detail in the main text.

Specific comments:

Title: I suggest the title "Study on downstream migration routes of atlantic salmon smolts at Fosstveit hydropower plant in Norway"

Line 94: Is the HPP really located 6 km upstream of the river mouth? According to the scale in Fig. 1a, it looks more like 20 km.

Line 95: Please state the design discharge Q_d of the HPP.

Line 98: Is it a conventional trash rack with vertical bars? Please specify.

Line 104: You mention a velocity of 0.5 m/s. What does it relate to? Is it the normal velocity at the trash rack ($v_n = \text{design discharge} / \text{rack area} = Q_d / A_r$)?

Line 112: "...the old river stretch..." Please use the term "residual flow stretch".

Lines 110 to 119: Please add the specific locations of all individual monitoring elements into Fig. 1. To me, Fig. 1a is not of too much information as the major information of that figure, the distance to the river mouth, is already stated in the main text. I suggest to skip it and instead to increase the figure size of both remaining subfigures 1b (mid scale) and 1c (small scale view of the HPP). I suggest to slightly increase the shown perimeter of both figures. One showing the general HPP layout with forebay, residual flow stretch, head race tunnel, power house, receiving river. The other one showing the individual HPP elements (please include symbols for PIT antennas and RSTs in addition to the arrows). Please add arrows for the flow direction.

Line 139: To assess the attraction efficiency of alternative migration routes it is crucial to know the discharge distribution between turbines, floodgates and the trash gate. It is important to know how far the trash gate effect reaches into the forebay. The flow velocities and opening dimensions finally determine if fish avoid the specific path.

Line 147: Which are the other surface gates? Please specify.

Line 154: 16 out of 444 fish were detected dead, right? So you can add a statement on the observed turbine survival rate.

Line 158: "Only a small fraction..." Please add specific numbers.

Line 164: "...other leading structures, ..." please write "guidance structures"

Line 172: 20 day delay. Is this related to the fish that did not migrate downstream until trash gate opening?

Line 184: "...are willing to postpone..." That's speculation. Maybe the fish can't decide for themselves as they don't find the migration path.

Line 193: Yes, the trash rack will act as a physical barrier for larger fish, and in addition for some (also smaller) fish as a behavioral barrier.

Figure 3: "Bypass migrants" There is no bypass. Please specify.

Figure 4: I suggest to shift the column "not encountered downstream" to the very left of the graph in order to match importance of the data.

Table 1 and Table 2: Please discuss the table contents in more detail in the main text, as described above.

Decision letter (RSOS-190989.R0)

13-Aug-2019

Dear Mr Haraldstad,

The Subject Editor assigned to your paper ("Migratory passage structures at hydropower plants as potential physiological and behavioural selective agents") has now received comments from reviewers. We would like you to revise your paper in accordance with the referee and Associate Editor suggestions which can be found below (not including confidential reports to the Editor). Please note this decision does not guarantee eventual acceptance.

Please submit a copy of your revised paper before 05-Sep-2019. Please note that the revision deadline will expire at 00.00am on this date. If we do not hear from you within this time then it will be assumed that the paper has been withdrawn. In exceptional circumstances, extensions may be possible if agreed with the Editorial Office in advance. We do not allow multiple rounds of revision so we urge you to make every effort to fully address all of the comments at this stage. If deemed necessary by the Editors, your manuscript will be sent back to one or more of the original reviewers for assessment. If the original reviewers are not available we may invite new reviewers.

When submitting your revised manuscript, you must respond to the comments made by the referees and upload a file "Response to Referees" in "Section 6 - File Upload". Please use this to document how you have responded to each of the comments, and the adjustments you have made. In order to expedite the processing of the revised manuscript, please be as specific as possible in your response.

- Ethics statement

- Data accessibility

<http://datadryad.org/submit?journalID=RSOS&manu=RSOS-190989>

- Competing interests

- Authors' contributions

All submissions, other than those with a single author, must include an Authors' Contributions section which individually lists the specific contribution of each author. The list of Authors

should meet all of the following criteria; 1) substantial contributions to conception and design, or acquisition of data, or analysis and interpretation of data; 2) drafting the article or revising it critically for important intellectual content; and 3) final approval of the version to be published.

- Acknowledgements

- Funding statement

on behalf of Dr Michael Tobler (Associate Editor) and Kevin Padian (Subject Editor)
openscience@royalsociety.org

Editor Comments:

Thank you for your efforts at revising. The reviewers are generally well disposed toward the manuscript but offer some further comments, as well as the suggestion to have the manuscript edited by a native speaker of English, with which I agree. It is not bad at all but needs to be cleaned up a bit. This and the reviewers' comments will need to be fully addressed in the next resubmission; we are not able to entertain another round of revision. And please keep the title you have, because it has more informative key words than the one suggested by one of our reviewers. Best wishes.

Reviewer comments to Author:

Reviewer: 2

Comments to the Author(s)

Review of RSOS-190989: Migratory passage structures at hydropower plants as potential physiological and behavioral selective agents.

The revisions the authors have incorporated have improved the manuscript greatly. The new

analyses are clear and the figures tell the story really well. I have some additional minor comments for the authors to consider, and overall I thought the Discussion was very solid. I did not spend much time on editorial-style comments because, as I pointed out in my last review, I think the manuscript needs the attention of an English language service to fix numerous sentence structure issues.

Abstract L18: Provide a specific number of increase here. How much did the proportion increase?

Introduction P5, L34: I don't think this paragraph is necessary. These points are good ones, but could be incorporated elsewhere with citations. Authors generally say "the river", but are they referring to rivers in general or a specific river?

Methods P7, L35: Provide a citation for these previous findings of incision healing.

Methods P7, L53: What concession? Please clarify for readers unfamiliar with this system.

Methods P8, L5: Figure 1 only depicts two traps. Put an arrow to the "rivermouth" trap in the top panel.

Methods P8, L33: Not clear what "recapture" is referring to here. If these are antenna detections then this needs clarified. When referring to physical captures, recapture would be appropriate, and for PIT antenna detections, detections would be appropriate. If the authors combine this two types of data then something like "reencounters" might be more accurate. Also, report a total sample size for this detection probability. Would it be N=250 individuals?

Methods P9, L3: divide by the total number of tagged smolts?

Results P9, L46: Report plus/minus 14.5 directly after "139.0 mm" and just put "SD" in parentheses.

Results P10, L50: Difficult to interpret what "not very different" means? It would be helpful to present numbers here so readers can assess quantitative differences.

Results P10, L51: Need to explain what this interaction term means. The letters mean nothing to readers because this term isn't explained until Table 3.

Results P10, L59: I suggest the authors provide some interpretation of this result and the difference between early- and late-release cohorts. This is likely due to late-release fish still searching for a downstream route, while early-release fish have been in the forebay for some time.

Discussion P11, L18: Something really striking to me was how little fish used the fish passage for downstream migration. I think the authors should make a bigger point of this and call attention here in the Discussion. This could be done in a sentence or two. It's often assumed that construction of a fish passage automatically restores functional connectivity, few studies assess fish passage structures, and fewer yet assess downstream migration through fish passages. The data presented in this study are pretty striking that fish passages far away from turbine channels, where I presume many are often built on hydro-power facilities, are completely ineffective for downstream migration at least for this life stage and species. This is really good information for managers to have.

Discussion P11, L36: could be occurring.

Discussion P12, L30: fish could also have died.

Discussion P12, L52: So, a behavioral barrier might exist?

Discussion P13, L3-37: This is a good paragraph.

Figure 4 and 5: Report units for "Time to migration".

Table 3: This is a minor formatting issue, but the "Term" column appears to be vertically justified to the top of the cell, while the statistical parameters are bottom justified. This makes it difficult to quickly skim across rows. Please change.

Reviewer: 3

Comments to the Author(s)

Manuscript content summary:

The authors present a study on downstream migration routes of atlantic salmon at hydropower plant (HPP) Fosstveit in Norway. Within 23 days (April 30 – May 21, 2010) a total of 923 salmon smolts were caught, PIT tagged, and released upstream of the HPP. Four possible downstream migration routes exist (1) the turbine, (2) floodgates, (3) a fish ladder, and (4) a trash gate. The turbine outlet was individually monitored with a rotary screw trap equipped with nets. The remaining 3 migration routes were monitored conjointly with PIT-antennas in the residual flow reach downstream of the HPP. On May 20 and 21 the trash gate was opened, leading to altered flow conditions in the HPP forebay, such that the number of fish detected in the residual flow reach significantly increased on that 2 days.

General comments:

The manuscript is well written and the level of English is good. I consider the manuscript contents as very interesting and valuable for the research community given the below corrections and clarifications will be included.

To me, some important details such as the specific locations of the monitoring devices, the characteristics (total river discharge, turbine discharge, fish ladder discharge, floodgate discharge, floodgate dimensions, trash gate discharge, trash gate opening dimensions, ...) of all individual HPP elements and the exact numbers of detected fish are unclear and should be explained in greater detail in the main text.

Specific comments:

Title: I suggest the title "Study on downstream migration routes of atlantic salmon smolts at Fosstveit hydropower plant in Norway"

Line 94: Is the HPP really located 6 km upstream of the river mouth? According to the scale in Fig. 1a, it looks more like 20 km.

Line 95: Please state the design discharge Q_d of the HPP.

Line 98: Is it a conventional trash rack with vertical bars? Please specify.

Line 104: You mention a velocity of 0.5 m/s. What does it relate to? Is it the normal velocity at the trash rack ($v_n = \text{design discharge} / \text{rack area} = Q_d / A_r$)?

Line 112: "...the old river stretch..." Please use the term "residual flow stretch".

Lines 110 to 119: Please add the specific locations of all individual monitoring elements into Fig. 1. To me, Fig. 1a is not of too much information as the major information of that figure, the distance to the river mouth, is already stated in the main text. I suggest to skip it and instead to increase the figure size of both remaining subfigures 1b (mid scale) and 1c (small scale view of the HPP). I suggest to slightly increase the shown perimeter of both figures. One showing the general HPP layout with forebay, residual flow stretch, head race tunnel, power house, receiving river. The other one showing the individual HPP elements (please include symbols for PIT antennas and RSTs in addition to the arrows). Please add arrows for the flow direction.

Line 139: To assess the attraction efficiency of alternative migration routes it is crucial to know the discharge distribution between turbines, floodgates and the trash gate. It is important to know how far the trash gate effect reaches into the forebay. The flow velocities and opening dimensions finally determine if fish avoid the specific path.

Line 147: Which are the other surface gates? Please specify.

Line 154: 16 out of 444 fish were detected dead, right? So you can add a statement on the observed turbine survival rate.

Line 158: "Only a small fraction..." Please add specific numbers.

Line 164: "...other leading structures, ..." please write "guidance structures"

Line 172: 20 day delay. Is this related to the fish that did not migrate downstream until trash gate opening?

Line 184: "...are willing to postpone..." That's speculation. Maybe the fish can't decide for themselves as they don't find the migration path.

Line 193: Yes, the trash rack will act as a physical barrier for larger fish, and in addition for some (also smaller) fish as a behavioral barrier.

Figure 3: "Bypass migrants" There is no bypass. Please specify.

Figure 4: I suggest to shift the column "not encountered downstream" to the very left of the graph in order to match importance of the data.

Table 1 and Table 2: Please discuss the table contents in more detail in the main text, as described above.

Author's Response to Decision Letter for (RSOS-190989.R0)

See Appendix B.

Decision letter (RSOS-190989.R1)

23-Oct-2019

Dear Mr Haraldstad,

I am pleased to inform you that your manuscript entitled "Migratory passage structures at hydropower plants as potential physiological and behavioural selective agents" is now accepted for publication in Royal Society Open Science.

Kind regards,
Lianne Parkhouse
Editorial Coordinator
openscience@royalsociety.org

on behalf of Dr Michael Tobler (Associate Editor) and Professor Kevin Padian (Subject Editor)
openscience@royalsociety.org

Appendix A

Dear Editor

We are pleased to resubmit a revised version of the manuscript RSOS-181741 entitled “Migratory passage structures at hydropower plants as potential physiological and behavioural selective agents” We appreciated the constructive criticisms from reviewer2 and appreciate that the reviewer has noticed an improvement in the resubmitted version. We addressed the new concerns as outlined below. Responses are marked with # and line numbers refer to the new manuscript uploaded at the RSOS website.

Unfortunately, it looks like Reviewer 3 has reviewed the first submitted version and not the resubmitted version. Nonetheless some of the concerns reviewer 3 address is relevant also for the resubmitted version and greatly improve the manuscript.

The new manuscript has been edited by a native speaker of English, as suggested by the reviewer and the Editor. We have not given specific locations of the changes.

Editor Comments:

Thank you for your efforts at revising. The reviewers are generally well disposed toward the manuscript but offer some further comments, as well as the suggestion to have the manuscript edited by a native speaker of English, with which I agree. It is not bad at all but needs to be cleaned up a bit. This and the reviewers' comments will need to be fully addressed in the next resubmission; we are not able to entertain another round of revision. And please keep the title you have, because it has more informative key words than the one suggested by one of our reviewers. Best wishes.

Reviewer comments to Author:

Reviewer: 2

Comments to the Author(s)

Review of RSOS-190989: Migratory passage structures at hydropower plants as potential physiological and behavioral selective agents.

The revisions the authors have incorporated have improved the manuscript greatly. The new analyses are clear and the figures tell the story really well. I have some additional minor comments for the authors to consider, and overall I thought the Discussion was very solid. I did not spend much time on editorial-style comments because, as I pointed out in my last review, I think the manuscript needs the attention of an English language service to fix numerous sentence structure issues.

Abstract L18: Provide a specific number of increase here. How much did the proportion increase?

corrected: The proportion of fish using the bypasses increased from 1% to 34% when water was released in surface gates closer to the turbine intake.

Introduction P5, L34: I don't think this paragraph is necessary. These points are good ones, but could be incorporated elsewhere with citations. Authors generally say “the river”, but are they referring to rivers in general or a specific river?

#This paragraph was added on request from a previous reviewer, and we suggest keeping it. We are referring to rivers in general, corrected.

Methods P7, L35: Provide a citation for these previous findings of incision healing.

#corrected

Methods P7, L53: What concession? Please clarify for readers unfamiliar with this system.

#corrected; A concession is needed to construct and operate a Norwegian hydropower plant and include site specific compensation measures to mitigate possible damage caused on the environment.

Methods P8, L5: Figure 1 only depicts two traps. Put an arrow to the “rivermouth” trap in the top panel.

#corrected

Methods P8, L33: Not clear what “recapture” is referring to here. If these are antenna detections then this needs clarified. When referring to physical captures, recapture would be appropriate, and for PIT antenna detections, detections would be appropriate. If the authors combine this two types of data then something like “reencounters” might be more accurate. Also, report a total sample size for this detection probability. Would it be N=250 individuals?

#Corrected. “Recaptures” changed to “detections” and total sample size was N=50, 10 individuals for each trail.

Methods P9, L3: divide by the total number of tagged smolts?

#No, total number of tagged smolts would introduce a bias due to the accumulation of smolts in the forebay and a part of the smolts have left the forebay through the turbine or flood gates in the days before. We argue that the daily fish guidance efficiency is the interesting metric here which is based on book-keeping of individuals available for migration at each day. We describe this in detail in the following sentence.

Results P9, L46: Report plus/minus 14.5 directly after “139.0 mm” and just put “SD” in parentheses.

#Corrected.

Results P10, L50: Difficult to interpret what “not very different” means? It would be helpful to present numbers here so readers can assess quantitative differences.

We have rephrased this sentence slightly and refer to the figure instead of listing the respective migration probabilities. Our point is that the release-cohorts show very similar migration probability trajectories for the before trash-gate migrants.

The new sentence: “Even though start day had a significant effect on migration probability, the predicted migration probability trajectories were not very different among release cohorts for the before-trash gate opening migrants (Figure 4)”

Results P10, L51: Need to explain what this interaction term means. The letters mean nothing to readers because this term isn’t explained until Table 3.

We agree and have rephrased this section to clarify what these interactions are and that they mainly affect the trash-gate migrants. The abbreviation letters have also been explained in the methods section to make them more recognizable.

*New phrasing: "However, because route was involved in significant interactions with both before/after opening trash gate (i.e., route*BA_TG) and with start day (i.e., route*start), this resulted in a substantial cohort effect for the trash gate migrants. In particular, early-release trash-gate migrant cohorts had high initial migration probabilities (typically > 0.7) at the opening day of the trash gate, but with relatively gentle response slope as time progressed (Figure 4). Later release trash-gate migrant cohorts had lower initial migration probabilities (~0.5) that rapidly progressed to cumulated migration probability of 1."*

Results P10, L59: I suggest the authors provide some interpretation of this result and the difference between early- and late-release cohorts. This is likely due to late-release fish still searching for a downstream route, while early-release fish have been in the forebay for some time.

#Agree, this is a topic for further studies, though we feel that this might be too much of a speculation to address based on our data.

Discussion P11, L18: Something really striking to me was how little fish used the fish passage for downstream migration. I think the authors should make a bigger point of this and call attention here in the Discussion. This could be done in a sentence or two. It's often assumed that construction of a fish passage automatically restores functional connectivity, few studies assess fish passage structures, and fewer yet assess downstream migration through fish passages. The data presented in this study are pretty striking that fish passages far away from turbine channels, where I presume many are often built on hydro-power facilities, are completely ineffective for downstream migration at least for this life stage and species. This is really good information for managers to have.

#Agree! Nice suggestion added in first two paragraphs of the discussion.

Discussion P11, L36: could be occurring.

#added

Discussion P12, L30: fish could also have died.

#That's true, added.

Discussion P12, L52: So, a behavioral barrier might exist?

#Yes

Discussion P13, L3-37: This is a good paragraph.

#Thanks

Figure 4 and 5: Report units for "Time to migration".

#Corrected

Table 3: This is a minor formatting issue, but the “Term” column appears to be vertically justified to the top of the cell, while the statistical parameters are bottom justified. This makes it difficult to quickly skim across rows. Please change.

#Corrected.

Reviewer: 3

Comments to the Author(s)

Manuscript content summary:

The authors present a study on downstream migration routes of atlantic salmon at hydropower plant (HPP) Fosstveit in Norway. Within 23 days (April 30 – May 21, 2010) a total of 923 salmon smolts were caught, PIT tagged, and released upstream of the HPP. Four possible downstream migration routes exist (1) the turbine, (2) floodgates, (3) a fish ladder, and (4) a trash gate. The turbine outlet was individually monitored with a rotary screw trap equipped with nets. The remaining 3 migration routes were monitored conjointly with PIT-antennas in the residual flow reach downstream of the HPP. On May 20 and 21 the trash gate was opened, leading to altered flow conditions in the HPP forebay, such that the number of fish detected in the residual flow reach significantly increased on that 2 days.

General comments:

The manuscript is well written and the level of English is good. I consider the manuscript contents as very interesting and valuable for the research community given the below corrections and clarifications will be included.

To me, some important details such as the specific locations of the monitoring devices, the characteristics (total river discharge, turbine discharge, fish ladder discharge, floodgate discharge, floodgate dimensions, trash gate discharge, trash gate opening dimensions, ...) of all individual HPP elements and the exact numbers of detected fish are unclear and should be explained in greater detail in the main text.

Specific comments:

Title: I suggest the title “Study on downstream migration routes of atlantic salmon smolts at Fosstveit hydropower plant in Norway”

Not a bad suggestion, but Editor preferred our title

p. Line 94: Is the HPP really located 6 km upstream of the river mouth? According to the scale in Fig. 1a, it looks more like 20 km.

#Yes, there is a large fjord (saltwater) that might be a bit confusing. So, you are correct regarding the distance between the HEP and the fjord mouth. River mouth is added in the upper panel. In a later comment you suggest skipping the upper panel, but we decide keeping it since it now contains more information (i.e. River mouth, RST and PIT-antenna).

Line 95: Please state the design discharge Q_d of the HPP.

We are not sure what you mean here. The hydropower plant maximum discharge is $16\text{ m}^3/\text{s}$ (Q_{max}), its written in the end of this paragraph. The discharge during the study period you can find in table 1 (3.7-4.7 m^3/s).

Line 98: Is it a conventional trash rack with vertical bars? Please specify.

#Yes, information added.

Line 104: You mention a velocity of 0.5 m/s. What does it relate to? Is it the normal velocity at the trash rack ($v_n = \text{design discharge} / \text{rack area} = Q_d / A_r$)?

Yes, this is the maximum powerplant discharge/rack area. It should be 0.64 and not 0.5. This error has been corrected. We are aware that water velocities may vary through the water column and that our calculation is not accurate in this regard. Unfortunately, we do not have more detailed measurements. Even so, the velocities are by far lower than reported swimming capacities of salmon smolts in the study by Peake and McKinley (see reference list).

Line 112: "...the old river stretch..." Please use the term "residual flow stretch".

#Corrected

Lines 110 to 119: Please add the specific locations of all individual monitoring elements into Fig. 1. To me, Fig. 1a is not of too much information as the major information of that figure, the distance to the river mouth, is already stated in the main text. I suggest to skip it and instead to increase the figure size of both remaining subfigures 1b (mid scale) and 1c (small scale view of the HPP). I suggest to slightly increase the shown perimeter of both figures. One showing the general HPP layout with forebay, residual flow stretch, head race tunnel, power house, receiving river. The other one showing the individual HPP elements (please include symbols for PIT antennas and RSTs in addition to the arrows). Please add arrows for the flow direction.

#Corrected.

Line 139: To assess the attraction efficiency of alternative migration routes it is crucial to know the discharge distribution between turbines, floodgates and the trash gate. It is important to know how far the trash gate effect reaches into the forebay. The flow velocities and opening dimensions finally determine if fish avoid the specific path.

Agree, the water velocity in front of the gates is probably an important factor affecting where fish go when they arrive at the dam. Unfortunately, we don't have such data. The opening dimensions of the different gates are given in Table 1.

Line 147: Which are the other surface gates? Please specify.

#The result section has been rewritten accordingly.

Line 154: 16 out of 444 fish were detected dead, right? So you can add a statement on the observed turbine survival rate.

We prefer not to, due to the uncertainties related to the catchability of dead fish. Fish may suffer lethal damages from the turbine, avoid our RST in the tailrace, only to be detected in the pit-antenna downstream (i.e. wrongly assessed as alive). We are working on exact estimates of turbine survival and post turbine survival at this particular site that will be presented in another paper in the future.

Line 158: "Only a small fraction..." Please add specific numbers.

corrected: "In this study only 22 out of 921 tagged Atlantic salmon smolts used the floodgates and the fish ladder during the initial..."

Line 164: "...other leading structures, ..." please write "guidance structures"

corrected

Line 172: 20 day delay. Is this related to the fish that did not migrate downstream until trash gate opening?

Yes, that is correct.

Line 184: "...are willing to postpone..." That's speculation. Maybe the fish can't decide for themselves as they don't find the migration path.

Agree, changed to "...part of the population might postpone migration..."

Line 193: Yess, the trash rack will act as a physical barrier for larger fish, and in addition for some (also smaller) fish as a behavioral barrier.

Agree

Figure 3: "Bypass migrants" There is no bypass. Please specify.

#Please see the new figures in the resubmitted version.

Figure 4: I suggest to shift the column "not encountered downstream" to the very left of the graph in order to match importance of the data.

#Please see the new figures in resubmitted version.

Table 1 and Table 2: Please discuss the table contents in more detail in the main text, as described above.

#Please note that Table 2 is changed in resubmitted version.

Appendix B

Dear Editor

We are pleased to resubmit a revised version of the manuscript RSOS-181741 entitled “Migratory passage structures at hydropower plants as potential physiological and behavioural selective agents” We appreciated the constructive criticisms from reviewer2 and appreciate that the reviewer has noticed an improvement in the resubmitted version. We addressed the new concerns as outlined below. Responses are marked with # and line numbers refer to the new manuscript uploaded at the RSOS website.

Unfortunately, it looks like Reviewer 3 has reviewed the first submitted version and not the resubmitted version. Nonetheless some of the concerns reviewer 3 address is relevant also for the resubmitted version and greatly improve the manuscript.

The new manuscript has been edited by a native speaker of English, as suggested by the reviewer and the Editor. We have not given specific locations of the changes.

Editor Comments:

Thank you for your efforts at revising. The reviewers are generally well disposed toward the manuscript but offer some further comments, as well as the suggestion to have the manuscript edited by a native speaker of English, with which I agree. It is not bad at all but needs to be cleaned up a bit. This and the reviewers' comments will need to be fully addressed in the next resubmission; we are not able to entertain another round of revision. And please keep the title you have, because it has more informative key words than the one suggested by one of our reviewers. Best wishes.

Reviewer comments to Author:

Reviewer: 2

Comments to the Author(s)

Review of RSOS-190989: Migratory passage structures at hydropower plants as potential physiological and behavioral selective agents.

The revisions the authors have incorporated have improved the manuscript greatly. The new analyses are clear and the figures tell the story really well. I have some additional minor comments for the authors to consider, and overall I thought the Discussion was very solid. I did not spend much time on editorial-style comments because, as I pointed out in my last review, I think the manuscript needs the attention of an English language service to fix numerous sentence structure issues.

Abstract L18: Provide a specific number of increase here. How much did the proportion increase?

corrected: The proportion of fish using the bypasses increased from 1% to 34% when water was released in surface gates closer to the turbine intake.

Introduction P5, L34: I don't think this paragraph is necessary. These points are good ones, but could be incorporated elsewhere with citations. Authors generally say “the river”, but are they referring to rivers in general or a specific river?

#This paragraph was added on request from a previous reviewer, and we suggest keeping it. We are referring to rivers in general, corrected.

Methods P7, L35: Provide a citation for these previous findings of incision healing.

#corrected

Methods P7, L53: What concession? Please clarify for readers unfamiliar with this system.

#corrected; A concession is needed to construct and operate a Norwegian hydropower plant and include site specific compensation measures to mitigate possible damage caused on the environment.

Methods P8, L5: Figure 1 only depicts two traps. Put an arrow to the “rivermouth” trap in the top panel.

#corrected

Methods P8, L33: Not clear what “recapture” is referring to here. If these are antenna detections then this needs clarified. When referring to physical captures, recapture would be appropriate, and for PIT antenna detections, detections would be appropriate. If the authors combine this two types of data then something like “reencounters” might be more accurate. Also, report a total sample size for this detection probability. Would it be N=250 individuals?

#Corrected. “Recaptures” changed to “detections” and total sample size was N=50, 10 individuals for each trail.

Methods P9, L3: divide by the total number of tagged smolts?

#No, total number of tagged smolts would introduce a bias due to the accumulation of smolts in the forebay and a part of the smolts have left the forebay through the turbine or flood gates in the days before. We argue that the daily fish guidance efficiency is the interesting metric here which is based on book-keeping of individuals available for migration at each day. We describe this in detail in the following sentence.

Results P9, L46: Report plus/minus 14.5 directly after “139.0 mm” and just put “SD” in parentheses.

#Corrected.

Results P10, L50: Difficult to interpret what “not very different” means? It would be helpful to present numbers here so readers can assess quantitative differences.

We have rephrased this sentence slightly and refer to the figure instead of listing the respective migration probabilities. Our point is that the release-cohorts show very similar migration probability trajectories for the before trash-gate migrants.

The new sentence: “Even though start day had a significant effect on migration probability, the predicted migration probability trajectories were not very different among release cohorts for the before-trash gate opening migrants (Figure 4)”

Results P10, L51: Need to explain what this interaction term means. The letters mean nothing to readers because this term isn’t explained until Table 3.

We agree and have rephrased this section to clarify what these interactions are and that they mainly affect the trash-gate migrants. The abbreviation letters have also been explained in the methods section to make them more recognizable.

*New phrasing: "However, because route was involved in significant interactions with both before/after opening trash gate (i.e., route*BA_TG) and with start day (i.e., route*start), this resulted in a substantial cohort effect for the trash gate migrants. In particular, early-release trash-gate migrant cohorts had high initial migration probabilities (typically > 0.7) at the opening day of the trash gate, but with relatively gentle response slope as time progressed (Figure 4). Later release trash-gate migrant cohorts had lower initial migration probabilities (~0.5) that rapidly progressed to cumulated migration probability of 1."*

Results P10, L59: I suggest the authors provide some interpretation of this result and the difference between early- and late-release cohorts. This is likely due to late-release fish still searching for a downstream route, while early-release fish have been in the forebay for some time.

#Agree, this is a topic for further studies, though we feel that this might be too much of a speculation to address based on our data.

Discussion P11, L18: Something really striking to me was how little fish used the fish passage for downstream migration. I think the authors should make a bigger point of this and call attention here in the Discussion. This could be done in a sentence or two. It's often assumed that construction of a fish passage automatically restores functional connectivity, few studies assess fish passage structures, and fewer yet assess downstream migration through fish passages. The data presented in this study are pretty striking that fish passages far away from turbine channels, where I presume many are often built on hydro-power facilities, are completely ineffective for downstream migration at least for this life stage and species. This is really good information for managers to have.

#Agree! Nice suggestion added in first two paragraphs of the discussion.

Discussion P11, L36: could be occurring.

#added

Discussion P12, L30: fish could also have died.

#That's true, added.

Discussion P12, L52: So, a behavioral barrier might exist?

#Yes

Discussion P13, L3-37: This is a good paragraph.

#Thanks

Figure 4 and 5: Report units for "Time to migration".

#Corrected

Table 3: This is a minor formatting issue, but the “Term” column appears to be vertically justified to the top of the cell, while the statistical parameters are bottom justified. This makes it difficult to quickly skim across rows. Please change.

#Corrected.

Reviewer: 3

Comments to the Author(s)

Manuscript content summary:

The authors present a study on downstream migration routes of atlantic salmon at hydropower plant (HPP) Fosstveit in Norway. Within 23 days (April 30 – May 21, 2010) a total of 923 salmon smolts were caught, PIT tagged, and released upstream of the HPP. Four possible downstream migration routes exist (1) the turbine, (2) floodgates, (3) a fish ladder, and (4) a trash gate. The turbine outlet was individually monitored with a rotary screw trap equipped with nets. The remaining 3 migration routes were monitored conjointly with PIT-antennas in the residual flow reach downstream of the HPP. On May 20 and 21 the trash gate was opened, leading to altered flow conditions in the HPP forebay, such that the number of fish detected in the residual flow reach significantly increased on that 2 days.

General comments:

The manuscript is well written and the level of English is good. I consider the manuscript contents as very interesting and valuable for the research community given the below corrections and clarifications will be included.

To me, some important details such as the specific locations of the monitoring devices, the characteristics (total river discharge, turbine discharge, fish ladder discharge, floodgate discharge, floodgate dimensions, trash gate discharge, trash gate opening dimensions, ...) of all individual HPP elements and the exact numbers of detected fish are unclear and should be explained in greater detail in the main text.

Specific comments:

Title: I suggest the title “Study on downstream migration routes of atlantic salmon smolts at Fosstveit hydropower plant in Norway”

Not a bad suggestion, but Editor preferred our title

p. Line 94: Is the HPP really located 6 km upstream of the river mouth? According to the scale in Fig. 1a, it looks more like 20 km.

#Yes, there is a large fjord (saltwater) that might be a bit confusing. So, you are correct regarding the distance between the HEP and the fjord mouth. River mouth is added in the upper panel. In a later comment you suggest skipping the upper panel, but we decide keeping it since it now contains more information (i.e. River mouth, RST and PIT-antenna).

Line 95: Please state the design discharge Q_d of the HPP.

We are not sure what you mean here. The hydropower plant maximum discharge is $16\text{ m}^3/\text{s}$ (Q_{max}), its written in the end of this paragraph. The discharge during the study period you can find in table 1 (3.7-4.7 m^3/s).

Line 98: Is it a conventional trash rack with vertical bars? Please specify.

#Yes, information added.

Line 104: You mention a velocity of 0.5 m/s. What does it relate to? Is it the normal velocity at the trash rack ($v_n = \text{design discharge} / \text{rack area} = Q_d / A_r$)?

Yes, this is the maximum powerplant discharge/rack area. It should be 0.64 and not 0.5. This error has been corrected. We are aware that water velocities may vary through the water column and that our calculation is not accurate in this regard. Unfortunately, we do not have more detailed measurements. Even so, the velocities are by far lower than reported swimming capacities of salmon smolts in the study by Peake and McKinley (see reference list).

Line 112: "...the old river stretch..." Please use the term "residual flow stretch".

#Corrected

Lines 110 to 119: Please add the specific locations of all individual monitoring elements into Fig. 1. To me, Fig. 1a is not of too much information as the major information of that figure, the distance to the river mouth, is already stated in the main text. I suggest to skip it and instead to increase the figure size of both remaining subfigures 1b (mid scale) and 1c (small scale view of the HPP). I suggest to slightly increase the shown perimeter of both figures. One showing the general HPP layout with forebay, residual flow stretch, head race tunnel, power house, receiving river. The other one showing the individual HPP elements (please include symbols for PIT antennas and RSTs in addition to the arrows). Please add arrows for the flow direction.

#Corrected.

Line 139: To assess the attraction efficiency of alternative migration routes it is crucial to know the discharge distribution between turbines, floodgates and the trash gate. It is important to know how far the trash gate effect reaches into the forebay. The flow velocities and opening dimensions finally determine if fish avoid the specific path.

Agree, the water velocity in front of the gates is probably an important factor affecting where fish go when they arrive at the dam. Unfortunately, we don't have such data. The opening dimensions of the different gates are given in Table 1.

Line 147: Which are the other surface gates? Please specify.

#The result section has been rewritten accordingly.

Line 154: 16 out of 444 fish were detected dead, right? So you can add a statement on the observed turbine survival rate.

We prefer not to, due to the uncertainties related to the catchability of dead fish. Fish may suffer lethal damages from the turbine, avoid our RST in the tailrace, only to be detected in the pit-antenna downstream (i.e. wrongly assessed as alive). We are working on exact estimates of turbine survival and post turbine survival at this particular site that will be presented in another paper in the future.

Line 158: “Only a small fraction...” Please add specific numbers.

corrected: “In this study only 22 out of 921 tagged Atlantic salmon smolts used the floodgates and the fish ladder during the initial...”

Line 164: “...other leading structures, ...” please write “guidance structures”

corrected

Line 172: 20 day delay. Is this related to the fish that did not migrate downstream until trash gate opening?

Yes, that is correct.

Line 184: “...are willing to postpone...” That’s speculation. Maybe the fish can’t decide for themselves as they don’t find the migration path.

Agree, changed to “...part of the population might postpone migration...”

Line 193: Yess, the trash rack will act as a physical barrier for larger fish, and in addition for some (also smaller) fish as a behavioral barrier.

Agree

Figure 3: “Bypass migrants” There is no bypass. Please specify.

#Please see the new figures in the resubmitted version.

Figure 4: I suggest to shift the column “not encountered downstream” to the very left of the graph in order to match importance of the data.

#Please see the new figures in resubmitted version.

Table 1 and Table 2: Please discuss the table contents in more detail in the main text, as described above.

#Please note that Table 2 is changed in resubmitted version.